# Implementation and validation of a supermodelling framework into CESM version 2.1.5

William E. Chapman [1], Francine Schevenhoven [2], Judith Berner [1], Noel Keenlyside [2,3], Ingo Bethke [2], Ping-Gin Chiu [2], Alok Gupta [3], and Jesse Nusbaumer [1]

[1]National Center for Atmospheric Research, Boulder, Co. USA
[2]Geophysical Institute and Bjerknes Centre for Climate Research, University of Bergen, Bergen, Norway
[3]Nansen Environmental and Remote Sensing Center, Bergen, Norway

**Correspondence:** William E. Chapman (wchapman@ucar.edu)

**Abstract.**

Here we present a research framework for the first atmosphere-connected supermodel using state-of-the-art atmospheric models. The Community Atmosphere Model (CAM) versions 5 and 6 exchange information interactively while running, a process known as supermodeling. The primary goal of this approach is to synchronize the models, allowing them to create a new dynamical system which can theoretically benefit from each component model, in part by increasing the dimensionality of the system.

In this study, we examine a single untrained supermodel where each model version is equally weighted in creating pseudo-observations. We demonstrate that the models synchronize well without decreased variability, particularly in storm track regions, across multiple timescales and for variables where no information has been exchanged. Synchronization is less pronounced in the tropics, and in regions of lesser synchronization we observe a decrease in high-frequency variability. Additionally, the low-frequency modes of variability (North Atlantic Oscillation and Pacific North American Pattern) are not degraded compared to the base models. For some variables, the mean bias is reduced compared to control simulations of each model version as well as the non-interactive ensemble mean.

## 1 Introduction

Climate models are essential for understanding and analyzing the complex dynamics of our Earth system. However, significant uncertainties remain, primarily due to the challenges in accurately parameterizing key processes and the biases inherent in different components of these models. These biases often exceed the projected climate change signals and the natural background variability that we aim to predict (Palmer and Stevens, 2019). Numerous options to improve climate representation are actively being explored, including enhancing subgrid physics, the incorporation of stochastic terms (e.g., Berner et al., 2008, 2012, 2017), utilizing machine-learned parameterizations and closures developed from observations or high-resolution model runs (e.g., Gregory et al., 2023; Chapman and Berner, 2024; Watt-Meyer et al., 2021; Bretherton et al., 2022), and increasing climate model resolution to directly resolve specific processes instead of parameterizing them (e.g., Judt, 2018; Palmer, 2014; Segura et al., 2025). Often these approaches effectively increase the dimensionality of the prediction system in

some form by adding additional degrees of freedom. However, these methods are challenging to develop and often too computationally intensive to practically implement. Despite these challenges, progress is crucial for improving climate predictions and informing policy decisions on climate adaptation and mitigation. Thus, alternative methods, which rely on the current generation of models, must be tested.

A simple approach to improving model representation is multi-model averaging, performed after individual models have been run. This method has been shown to reduce climate model biases in various applications (e.g., North American Multi-Model Ensemble, Coupled Model Intercomparison Project). These non-interacting ensembles (NIE) reduce errors by balancing the biases from multiple models. Moreover, advanced ensemble weighting schemes can further improve NIE effectiveness (e.g., Weigel et al., 2010; Tegegne et al., 2019). However, NIEs have limitations because they cannot combine model outputs in real-time, making them confined to the attractor space of each individual model. Additionally, biases that are shared across the individual models in an NIE cannot be corrected due to the linear nature of post-process averaging.

Supermodels are designed to address this limitation by creating a new synchronized dynamical system, which consists of the individual models interacting during run time by exchanging either state or tendency information. Since the models exchange information at runtime, the interactive ensemble is effectively of higher dimensionality. Taking advantage of model diversity compensates for individual model bias errors and allows more complex dynamical behavior. However, this is often evidenced in representations of localized structures, rather than in reductions in mean squared error (Duane and Shen, 2023). Supermodeling is a generalization of the interactive ensemble approach introduced by Kirtman and Shukla (2002), who coupled multiple realizations of the same atmospheric general circulation model to a single ocean general circulation model through averaging each models' air-sea fluxes. Since then a number of efforts have focused on linking increasingly complex models from low-dimensional simple models to models of intermediate complexity (van den Berge et al., 2011; Duane et al., 2009, 2018; Schevenhoven et al., 2023), and a framework for a state-of-the-art ocean-connected supermodel has been developed (Counillon et al., 2023).

Supermodels depend on two key principles: firstly, the synchronization of different models rooted in the concept of chaos synchronization in non-linear dynamical systems (Duane and Tribbia, 2001; Pecora et al., 1997); and secondly, the diversity among models can reflect the actual behavior of the target dynamical system. The models can either be directly linked to each other via a subset or all variables (e.g., van den Berge et al., 2011) or connected to their weighted average (e.g., Schevenhoven, 2021; Wiegerinck et al., 2013). The weighted average is also referred to as pseudo-observations, a terminology which will be explained below. A central point is that if the models are sufficiently synchronized the supermodel should not suffer from a decrease in variance. A decrease of variance in a particular region or variable is a sign of an insufficient synchronization and might require the exchange of more information between the interactive ensemble members (Wiegerinck et al., 2013).

To achieve optimal performance, a supermodel must be trained using data from the "truth" such as observations or a reference model. During the training phase, the supermodel interaction coefficients are optimized to formulate the supermodel with the best skill. Since only the interaction coefficients need to be learned, the training effort is substantially less than that used by modern machine learning approaches which learn the entire forward operator of a model (e.g., Watt-Meyer et al., 2023). Efficient methods have been developed to train supermodels (Schevenhoven and Selten, 2017) and have been shown in a

coupled model of intermediate complexity, SPEEDO, connected via the atmospheres only (Severijns and Hazeleger, 2010),
these training methods showed promising results (Schevenhoven et al., 2019), even when the observations were sparse and
noisy (Schevenhoven, 2021). As the goal of this manuscript is to describe the modeling framework itself, we will henceforth
focus on results from an untrained supermodel.

Exchanging information between models at runtime introduces substantial technical complexity and difficulties. Conceptually, each model has to be integrated forward for a number of model timesteps and then paused. Then, information between
the component models is exchanged. For simple models all model variables will be available on a single processor and the
exchange of information is straight forward. However, for more complex models which need to run on large distributed memory systems, this information sharing can be more difficult. Often, input/output files need to be written, combined, and each
component model has to be 'restarted' or 'resumed' after reading in exchanged information. This methodology is reminiscent
of the traditional data assimilation (DA) approach, which attempts to synchronize a numerical weather forecast (NWP) with the
observed atmospheric state. As such traditional data assimilation in NWP can be considered a special case of supermodeling.
This analogy explains why the exchanged information is also called "pseudo-observation".

Duane et al. (2006) recognized this analogy and suggested the use of traditional data assimilation tools (e.g., Du and Smith,
2017) to ingest the pseudo-observations into the components models. This approach was adopted by Counillon et al. (2023)
to successfully connect multiple ocean models. However, the overhead of writing and reading input/output files and restarting
the component models is computationally very inefficient and prohibits interaction at every timestep. Furthermore a workflow
manager is necessary to create the pseudo-observations and delay the restart until the latter are available, especially if one of
the component models is slower then the other. Additionally, while some models have pause/resume capability and can be
restarted quickly, others, like the one used in this study, take a relatively long time to re-initialize the model, which makes the
DA approach for creating a supermodel undesirable.

Here, we describe the technical details of the implementation of the so-far most complex supermodel in a heavily parallelized
High Performance Computing environment. It connects two versions of the Community Atmosphere Model (CAM), which is
the atmospheric component of the Community Earth System Model CESM (Danabasoglu et al., 2020). The advantage over
previous implementations come in through three new developments: 1) the availability of a newly developed Python-Fortran-
Bridge in CESM, 2) the adaptation of the existing nudging toolbox (Chapman and Berner, 2023) for our purposes, and 3) the
submission of multiple jobs through a single PBS or SLURM scheduler, which allows both component models to get into the
same queue.

A newly developed Python-Fortran-Bridge is now available in CESM which allows calls to python routines from the Fortran
executable at runtime. Called via the CESM-internal workflow, the python-calls take the role of the workflow manager, control
the generation of the pseudo-observations and effectively introduces pause/resume functionality into CESM.

The CESM nudging toolbox was used by Chapman and Berner (2024) to compare nudging tendencies to DA increments.
In the supermodeling context, its infrastructure can be easily adapted to facilitate nudging to the (weighted) model-average as
well as specifications for the interaction interval and which variables should be connected via Fortran namelist parameters.

While we present only first results using an unweighted, atmosphere-connected supermodel, we stress that the interoperability of the implementation will make extensions to include ocean-connections straight forward. It is also easy to add additional component models, as long as they are available on the same supercomputer. We also use the above-described training methods to optimize the performance of the supermodel, the results of which will be described in a forthcoming manuscript.

The manuscript is structured as follows: Section 2 describes the components models, verification datatsets and implementation of the supermodel, Section 3 discusses model synchronization and presents results supporting a successful implementation. Section 4 provides a discussion and concludes with the findings.

## 2 Implementation and Synchronization Methodology

### 2.1 Component Models and experiment setup

We combine different simulations of the Community Atmosphere Model (CAM), an atmospheric general circulation model (AGCM) developed at the National Center for Atmospheric Research with extensive community support. Our supermodel integrates CAM version 5 (CAM5; Neale et al., 2010) and CAM version 6 (CAM6; Bogenschutz et al., 2018; Gettelman et al., 2018), each incorporating different physics suites while using the same finite-volume (FV) dynamical core. CAM5 is released as the atmospheric component of CESM1 and CAM6 of CESM2, respectively.

The CAM5 simulation is run from the CAM6 code base with the CAM5 physics flag activated, which configures CAM to specifically use the physics schemes from CAM5.1 (CESM1.0.6). CAM5.1 treats stratiform cloud microphysics with a two-moment formulation (Morrison and Gettelman, 2008). The spatial distribution of shallow convection is simulated with a set of realistic plume dilution equations (Park and Bretherton, 2009). The ice cloud fraction scheme allows supersaturation via a modified relative humidity over ice and the inclusion of ice condensation amount (Gettelman et al., 2010). Descriptions for all other physics schemes (deep convection, PBL, radiation, etc.) can be found in Neale et al. (2010).

CAM6 uses the publicly released version of cam_cesm2_1_rel_60 from CESM2.1.5. Significant changes from CAM5 physics include substantial modifications to every atmospheric physics parameterization except for radiative transfer. The Cloud Layers Unified by Binormals (CLUBB, Golaz et al., 2002; Bogenschutz et al., 2013) scheme replaces CAM5 schemes for boundary layer turbulence, shallow convection, and cloud macrophysics. Additionally, an improved two-moment prognostic cloud microphysics (MG2 Gettelman and Morrison, 2015) was introduced between versions. The deep convection parameterization (Zhang and McFarlane, 1995) has been significantly retuned to increase sensitivity to convective inhibition. Both subgrid orographic drag calculation schemes have undergone substantial modifications. The orographic gravity wave scheme now incorporates topographic orientation (ridges) and low-level flow blocking effects. Finally, the previous parameterization of boundary layer form drag, known as turbulent mountain stress (TMS), has been replaced by the scheme of Beljaars et al. (2004).

While our supermodeling implementation utilizes interpolation routines to support different vertical and horizontal resolutions, we use here the resolution for which the atmospheric component models were scientifically released, namely a grid-size of 0.9°N x 1.25°E in the horizontal and 32 hybrid sigma-pressure levels up to 2.26 hPa in the vertical.

The model simulations followed the protocol of the Atmospheric Model Intercomparison Project (AMIP) and are forced by observed monthly sea surface temperatures and sea ice from 1979 to 2005 (26 years), with values linearly interpolated at each time step. The simulations also include prescribed evolutions of aerosol emissions and trace gas concentrations (including $CO_2$).

## 2.2 Validation Datasets

We verify the model against the $\sim 0.25°$ ERA5 reanalysis product (Hersbach et al., 2020) for all fields except precipitation, which is verified against the 1° NOAA GPCP product (Adler et al., 2003). For verification, the ERA5 product is bi-linearly interpolated to the native CAM grid prior to any metric calculation. The GPCP product is regridded to the native CAM grid using a conservative mapping method.

## 2.3 RMSE and Bias Calculation

As in the NCAR Atmospheric Modeling Working Group Diagnostic Package AMWG (2022), model error is calculated as the sum of the cosine- latitude weighted, root-mean-squared error (RMSE) of the spatial field after a seasonal, monthly, or daily mean has been computed. RMSE was used so that opposite-signed local biases do not cancel and erroneously inflate skill. Percent improvement is determined by first calculating global/regional RMSE and then calculating the percent change of RMSE compared to the reference.

## 2.4 Super Model Implementation

Our first attempt at implementing a supermodeling framework followed previous work (Counillon et al., 2023) and utilized a workflow manager, CYLC, together with tools from the data assimilation research testbed (DART Anderson et al., 2009) to restart the component models after their interaction via nudging to averaged output files. CESM-specific bottlenecks were 1) the time needed to re-initialize CESM after a restart, since the current CESM version does not have pause/resume capability and 2) re-entering the system queue after each interaction interval. Due to these inefficiencies completing a single year's simulation took approximately 24 hours (at 6-hour coupling), a cost that becomes untenable for multi-year simulations and made it impossible to increase the interaction frequency.

To overcome these difficulties, we devised a new custom workflow management system that eliminates the need for halting and re-initializing the model with each interchange of information between models by employing a `PAUSE/RESUME` mechanism. At the beginning of the physics timestep, the first component model outputs the model state variables—Zonal wind ($U$), Meridional wind ($V$), Temperature ($T$), and Specific Humidity ($Q$)—and initiates a model pause by writing a `PAUSE` file. Subsequently, CAM calls a Python script that waits for the second model to reach the beginning of its physics timestep and then combines the outputs from both models at the same timestamp. If the component model grids differ, Python interpolation routines are invoked to ensure consistency. Once the output has been processed, the Python script removes the `PAUSE` file, allowing the model to resume operation without the need for re-initialization or re-entering the queue. The implementation of

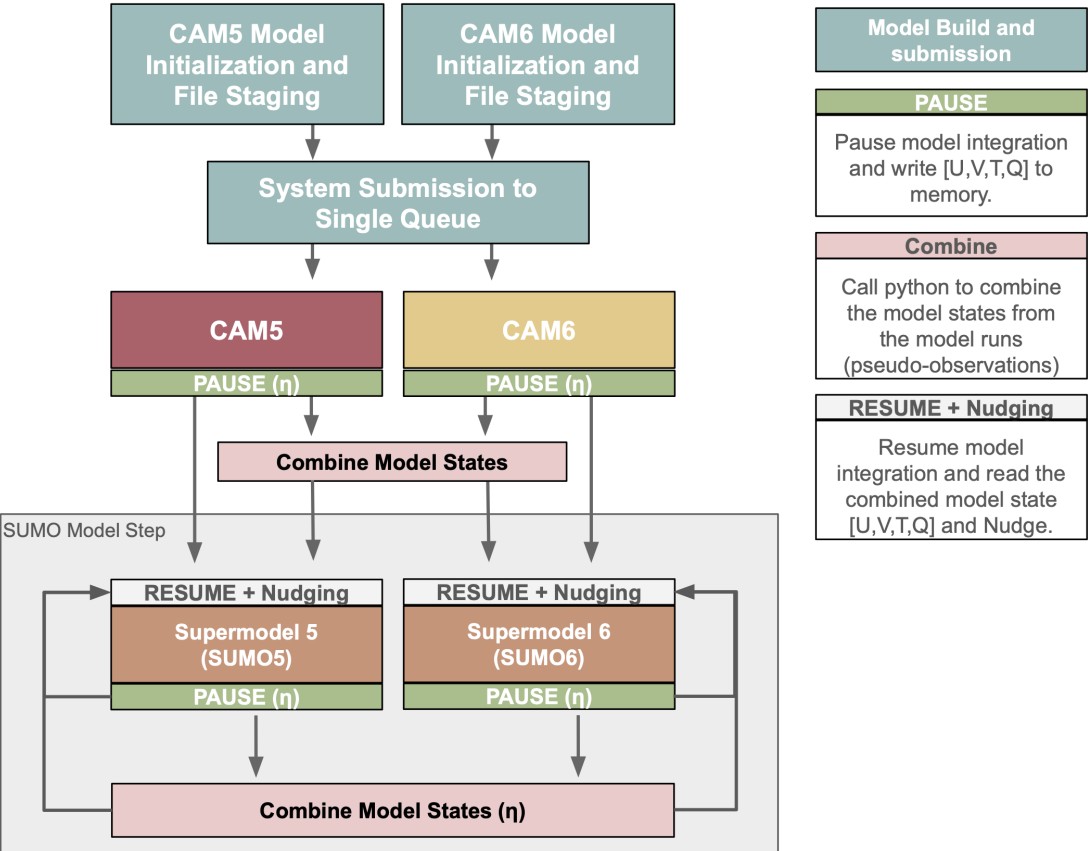

**Figure 1.** CAM5/6 Supermodel Workflow

efficient mpi-based communication between the models (i.e. standard coupling software) was beyond the scope of the study but is something that should be explored in future efforts.

A nudging tendency is then applied to each component model, following Equation (1), which nudges the model state toward the combined model state ($X_{\text{combined}}$) during the first timestep after the models resume running (e.g., Chapman and Berner, 2024). The user can set the nudging timescale ($\tau$); in this experiment, we use a relaxation timescale of 6 hours. Though we emphasize that the tendency is only applied at the first time-step after the combination (e.g., snapshot nudging).

$$\frac{dX}{dt} = F(X) + \frac{X_{\text{combined}} - X}{\tau} \tag{1}$$

where $X$ is the model state, $F(X)$ represents the model's internal tendencies, and $\tau$ is the nudging relaxation timescale.

To address the challenge of submitting multiple jobs through a single PBS/SLURM scheduler, we implemented a batch submission script that allows two model simulations to be launched concurrently while managing resource allocation efficiently.

Our approach ensures that both models utilize the available compute nodes without interfering with each other, thus avoiding scenarios where one model monopolizes the queue while the other remains pending.

Specifically, our submission script:

1. **Prepares model runs** by creating initialization files for both simulations.

2. **Defines model-specific execution settings**, including the number of processing elements required for each job.

3. **Partitions compute resources dynamically** by selecting appropriate node allocations from `$PBS_NODEFILE`, ensuring that both jobs receive the necessary resources without conflicts.

4. **Executes model runs in parallel** using background processes (`&`), allowing both jobs to start simultaneously while still being managed within a single job submission.

5. **Waits for all processes to complete** using `wait`, ensuring that computational resources are fully utilized before job completion.

This method ensures efficient job scheduling and mitigates the risk of asynchronous queuing delays, ultimately reducing computational time.

The resulting CAM5/CAM6 supermodel software workflow diagram is shown in Fig. 1. First, we provide scripts to build, compile, set namelist parameters, and stage necessary python and Fortran files. Then, all component models are submitted to the same submission queue using a single PBS or SLURM scheduler. The component models then run independently using their respective physics package until the first interaction timescale ($\eta$) is reached. The interaction timescale $\eta$ signifies the time after which the supermodels share information. At this point, each model writes output and is paused. Using a python-call from within the CESM workflow of one of the component models, the model states are subsequently averaged to create the combined state or pseudo-observation. The component models then resume, each being nudged towards the pseudo-observations. Once the next interaction timescale is reached, the models are paused and resume after their combined state has been computed. The last two steps are repeated until the desired simulation length is obtained (Fig. 1, grey box).

By adapting the previously developed nudging toolbox (Davis et al., 2022) for our purposes, we can easily set the interaction timescale, interacting variables, pseudo-observation file name, output path locations via namelist parameters.

The CAM5/CAM6 SuperModel, including the CESM Fortran-Python-bridge, supermodel module toolbox with namelist section, and scheduler scripts, are readily available via the GitHub repository. Currently, this system is deployed on two HPC platforms 1) the National Center for Atmospheric Research's Derecho Computer and 2) the Norwegian Research Infrastructure Services' machine, Betzy.

With these improvements, a one-year simulation is now accomplished in 7 hours, and is independent of system queuing time. Moreover, increasing the frequency of coupling does not significantly increase the wallclock time. We acknowledge that this is a significant slowdown from a CAM5/CAM6 simulation which can accomplish a year long simulation in $\sim 2.5$ hours with identical computational resources.

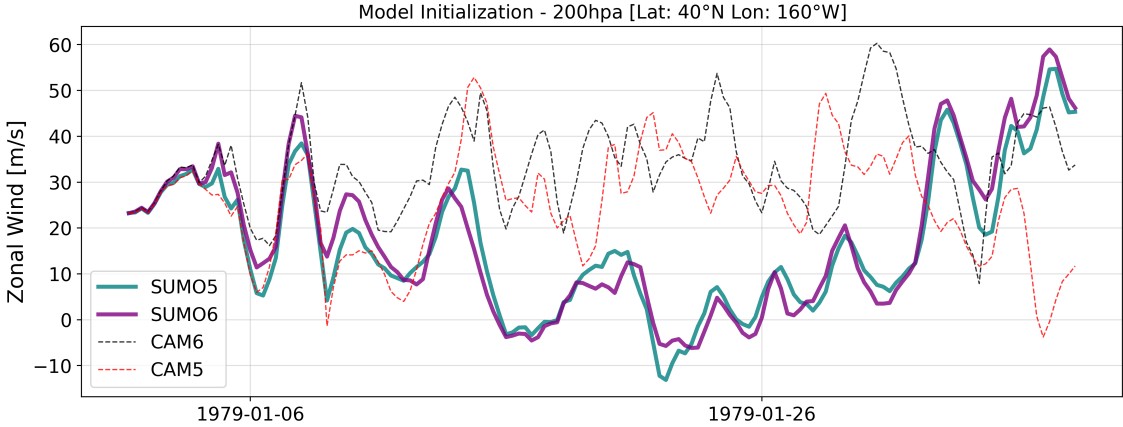

**Figure 2.** Experiments initialized from the same atmospheric state and integrated for one month. Showing: an independent CAM6 run (dashed black); an independent CAM5 run (dashed red); a supermodel which uses CAM5-physics (SUMO5; teal); a supermodel which uses CAM6-physics (SUMO6; purple) at location [Lat: 40N, Lon: 160W, Lev; 200 hPa].

## 3 Supermodel Results

200      We now demonstrate the synchronization and resulting mean state representation for the CAM5/CAM6 supermodel for the period 1979 through 2005. The supermodel uses an interaction timescale of $\eta = 6$ hours and employs snapshot nudging to the unweighted averaged state. In this implementation, the information in U, V, and T are exchanged and nudged while Q is left to evolve freely. We speculate that the main challenges with including specific humidity (Q) in the nudging process stem from the intrinsic properties of moisture in the atmosphere and its coupling with cloud and precipitation processes. This was done

205      because previous work indicated difficulty when adjusting specific humidity Q in CAM in both, nudging (e.g., Chapman and Berner, 2024) and full DA experiments (Raeder et al., 2021).

     We show results for four experiments: CAM5, CAM6, the supermodel which uses CAM5-physics, but is nudged to the combined state, SUMO5, and the supermodel which uses CAM6-physics, but is nudged to the combined state, SUMO6. We analyse the 6-hourly averaged prognostic state variables (U, V, T, Q, Surface Pressure (PS)) and standard CAM output which

210      is averaged monthly.

### 3.1 Synchronization

Figure 2 shows the zonal wind (U wind) at 200 hPa for four experiments started from the same initialization at a single model point ([40°N, 160°W]). The results indicate that while CAM5 and CAM6 vary independently, the two supermodel trajectories synchronize after ca. 15 day and the trajectories stay closely linked throughout the model run.

215      Figure 3 illustrates the anomaly Pearson correlation coefficients for four atmospheric variables: (U,V,T,Q) at a 6-hourly averaged temporal frequency. These correlations are computed between the two super models (SUMO5, SUMO6) and between

the individual CAM6 model and SUMO6 in model year 1979-1980. The analysis covers the 200hPa level at every grid point. To avoid anomalously high correlations for areas where the variability is largely driven by the annual cycle, we remove the 30-day centered rolling mean of the data. Figure 3 provides a quantitative measure of the degree of similarity between the two supermodel versions and the similarity between the one of the component models, CAM6, and the associated supermodel using also CAM6-physics. It highlights the effectiveness of the supermodeling approach in synchronizing the atmospheric state across different variables.

As to be expected there is no synchronization of the CAM6 and SUMO6 model states, as expressed by a low correlation coefficients of <0.15 for all almost all gridpoints (Fig. 3, right column). Correlations between the SUMO5 and SUMO6 experiments are much higher (Fig. 3, left column) indicating that the synchronization is evident not only across the component models, but also between the supermodels using different physics packages. Synchronization is strong in the U, V, and T fields poleward of $15°$, especially in the storm track regions. The Maritime continent region (Lat: $[15°S,15°N]$, Lon:$[60°E,200°W]$) displays the least amount of synchronization. Q displays less synchronization (Fig. 3g) with lowest correlation in the tropical belt, but still more than the correlation between CAM5 and CAM6. The supplemental material shows the same analysis for pressure levels of 750 hPa and 900 hPa (Fig. 1S and Fig. 2S). Generally, we see greater synchronization at of U,V, & T at higher pressure levels, while Q has a greater synchronization nearer to the surface, which could be a result of a similar sea surface temperature field between the two models.

If the component models are not sufficiently synchronized, the combined model state will exhibit diminished high-frequency variance compared to the individual models. This variance deflation occurs because the supermodel, representing a weighted average, tends to smooth out discrepancies between the models. As a result, the supermodel may lose critical variance, leading to reduced accuracy in capturing fine-scale variability. This issue is structurally related to the double penalty problem in modern machine learning for Numerical Weather Prediction (Brenowitz et al., 2024).

Poor synchronization between models, whether spatial or temporal, leads to an averaging effect that disproportionately smooths out high-frequency variations, dampening the system's true variability. Studies show that the less synchronized the models, the more the supermodel's variance is compromised by this effect, detracting from its ability to capture dynamic processes accurately (Counillon et al., 2023).

To examine the supermodel for signs of significant variance deflation, we compare histograms of 6-hourly averaged wind speed values (Fig. 4). We note that the CAM5 (red-dashed) and CAM6 (black-dashed) distributions are quite similar, this is likely do to the model tuning activity at NCAR prior to the model release. We detect a slight damping of the background winds near the mode of the distribution, but no degradation of the highest wind speeds. Overall, the difference between the component and super-models is minimal.

Even if the full fields do not suggest variance deflation, previous work using nudging (e.g., Chapman and Berner, 2024) suggests that any linear relaxation back to some sort of reference field is expected to reduce band-pass filtered variance. Hence we compute the standard deviation of the 12h to 5d band passed-filtered winds for U and V at 200 hPa (Figure 5). The zonal average (line), and zonal standard deviation (shading) are shown for SUMO6 (teal) and CAM6 (black) in the right hand column for the U and V winds (Fig. 5c & f, respectively). There is a significant damping of variability in this frequency band in areas

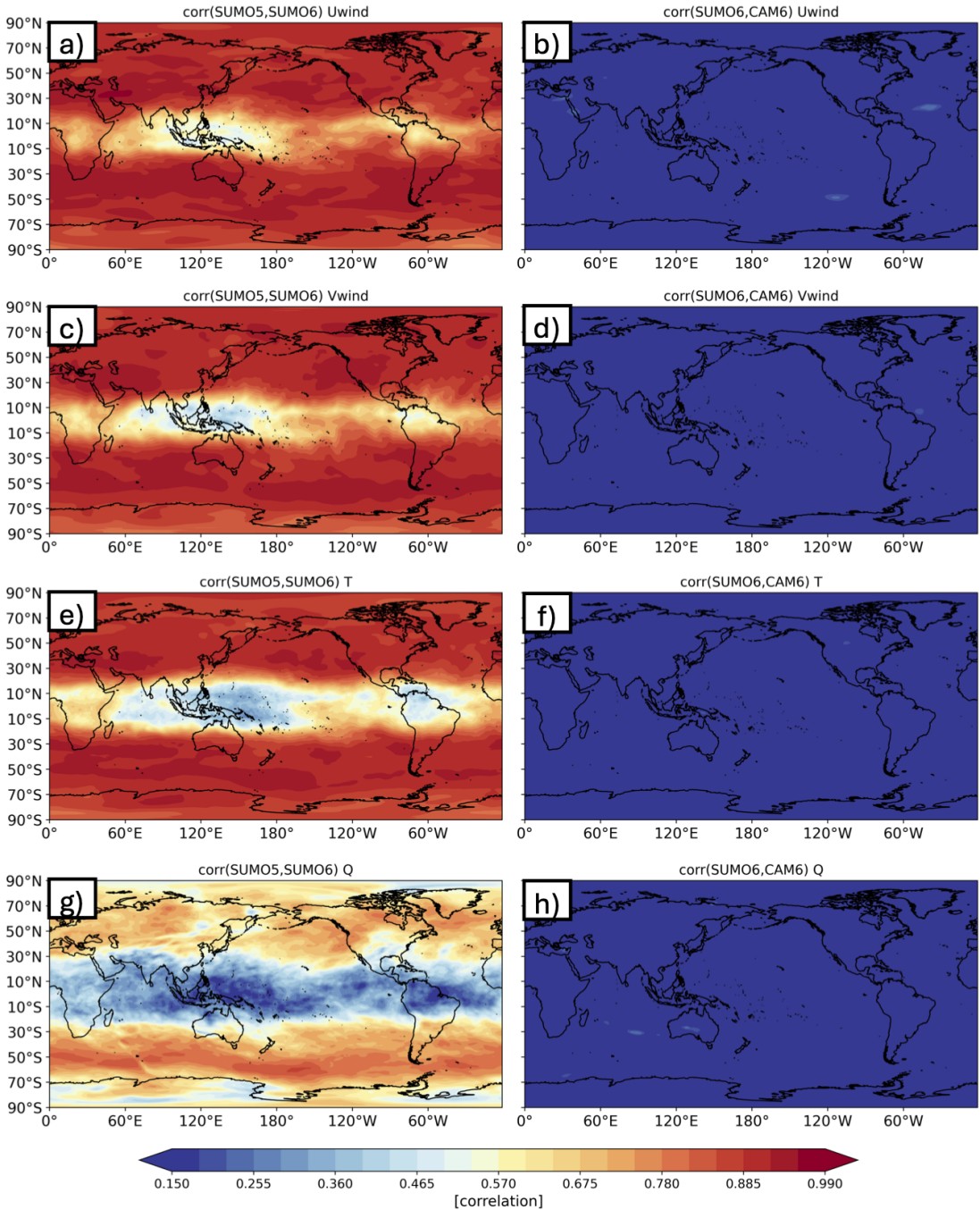

**Figure 3.** Anomaly correlation between the SUMO5 and SUMO6 s (left: a,c,e,g) and between the SUMO6 and CAM6 experiment (right: b,d,f,h). Shown are 6hourly-averaged model variables zonal wind (a,b), zonal wind (c,d), temperature (e,f), and specific humidity (g,h) at 200 hPa for the period spanning 1979-1980. Anomalies are computed by removing a 30-day centered mean at every timestep.

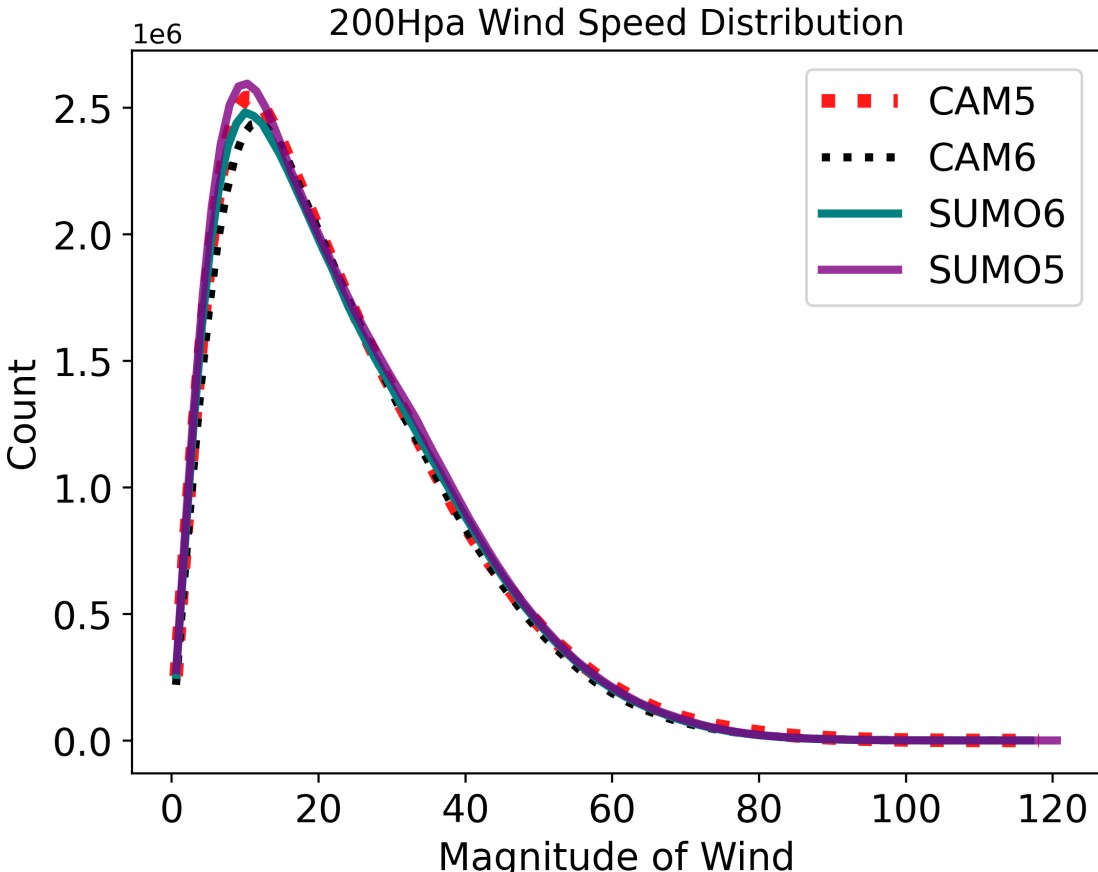

**Figure 4.** Histogram of all 6-hourly averaged wind speed values in model years 1979 at 200 hPa. Showing: an independent CAM6 run (black); an independent CAM5 run (red); the CAM5 supermodel (teal); CAM6 supermodel (purple).

where we find lower model synchronization like the tropics and the poles (Fig. 3a & c), especially over the maritime continent. In nudging studies, moving to an observations frequency of less than 6 hours seems to alleviate effects of damping (Davis et al., 2022), so we hypothesize that increasing the SUMO interaction frequency will be beneficial with regard to minimizing the damping of high frequency variability.

### 3.2 Low Frequency Modes of Variability

Evaluating the performance of an atmospheric model requires the adequate depiction of natural climate variability and significant low-frequency climate modes (e.g., Phillips et al., 2014). Intraseasonal variability arises from complex dynamical processes operating across multiple timescales, which subsequently influence downstream weather patterns (e.g., Branstator, 1992; Simmons et al., 1983; Wallace and Gutzler, 1981). The model's background climatology significantly influences this low-

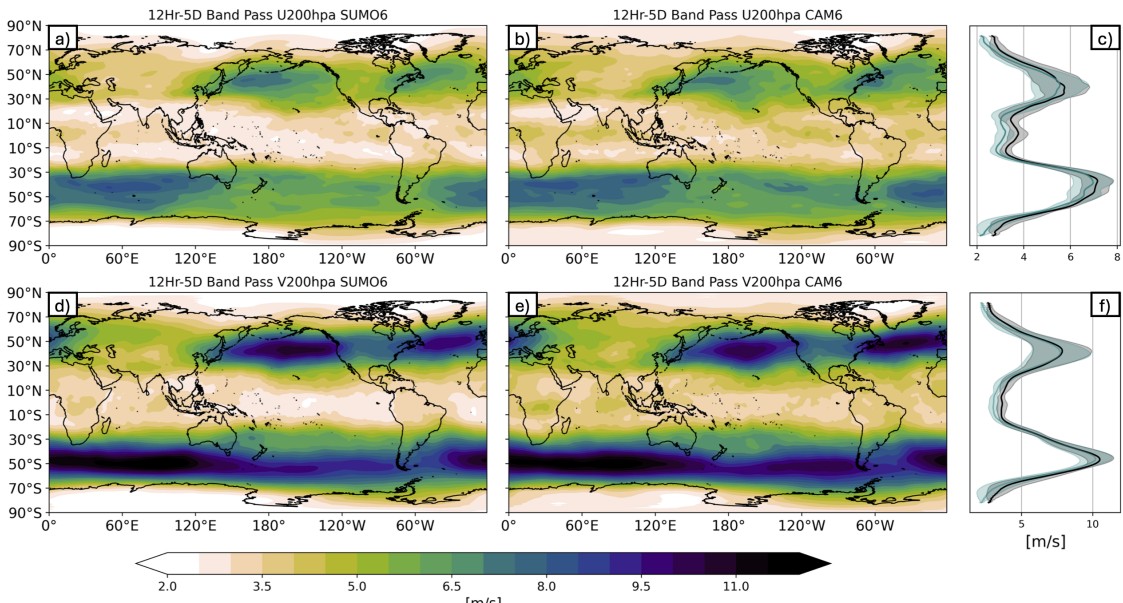

**Figure 5.** Standard deviation of 12 hour to 5 day band passed U (a,b) and V (d,e) winds for the SUMO6 (column I) and CAM6 (column II) model runs. And, the zonal average (line) and standard deviation (shading) of the the band passed winds for U (c) and V (f) for the SUMO6 (teal) and CAM6 (black) runs.

frequency variability, with several mechanisms proposed for its sustenance and growth. These mechanisms include the development of low-frequency anomalies due to instabilities in the zonally asymmetric midlatitude jet (e.g., Branstator, 1990, 1992; Frederiksen, 1983; Simmons et al., 1983), alterations in quasi-stationary eddies linked to changes in the zonal-mean flow (e.g., Branstator, 1984; Kang, 1990), tropical heating or orographic forcing (e.g., Hoskins and Karoly, 1981; Sardeshmukh
and Hoskins, 1988), and vorticity fluxes from high-frequency eddies (e.g., Branstator, 1992; Egger and Schilling, 1983; Lau, 1988; Ting and Lau, 1993). Accurately representing this low-frequency variability is vital for climate models, as the numerous interactions that contribute to an accurate depiction of low-frequency modes are indicative of the model's reliability.

To extract the leading patterns of variability we perform a empirical orthogonal function (EOF) decomposition on the monthly anomaly fields. The climatology is defined as the monthly mean from the full 26-year run or reanalysis product.
All EOF patterns are area weighted by the square root of the cosine (latitude) prior to decomposition. We express the orthogonal spatial field as the pointwise regression of each time series with a one-standard deviation change of the temporal principal component. The DJF Pacific - North American Pattern (PNA, Fig. 6), and North Atlantic Oscillation (NAO, Fig. 3S) are examined in detail. These patterns are defined as in Phillips et al. (2014) (NCAR's Climate Variability Diagnostic Package) as the leading mode of atmospheric variability in the region [20-85°N, 120°E-120°W] and [20-80°N, 90°W-40°E], respectively.
Twenty-six years and a single atmospheric realization is likely too short to adequately assess the spatial bias of either the PNA or the NAO (Deser et al., 2017). Therefore, we simply examine the patterns to show general loading locations (Fig. 6a-d) and

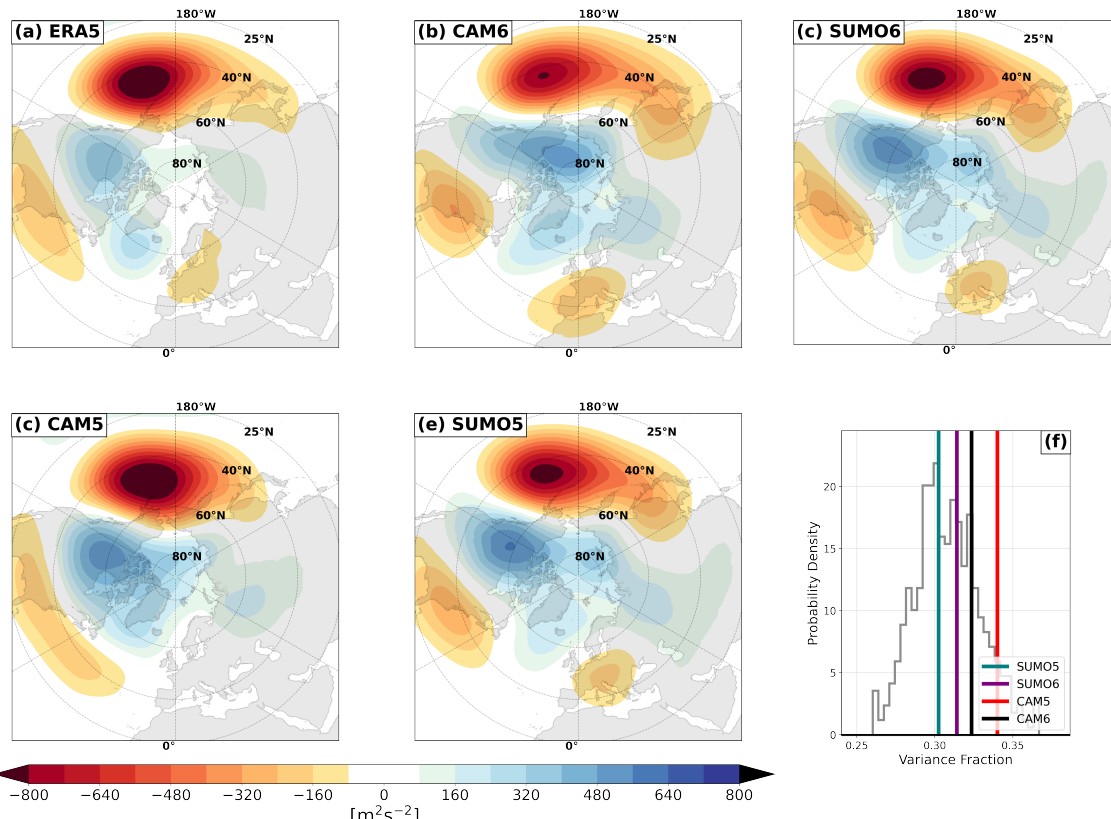

**Figure 6.** DJF 500 mb geopotential leading mode of variability over the region [NAO, 20-80°N, 90°W-40°E] ERA5 (a), CAM6 (b), CAM5 (c), and SUMO6 (d). Also, the explained variance in each model experiment (solid lines), and the bootstrapped spread of explained variance in the observations (e, grey histogram).

compare the representation of total variance to observations (Fig. 6e, grey histogram). To develop the histogram of explained variances we sub-sample the ERA5 observations into random twenty-six year chunks and bootstrap the EOF calculation 500 times (as in, Chapman and Berner, 2024). The PNA shows a classic stationary Rossby wave pattern spanning from the central
Pacific, across Canada and through Florida for every simulation. It is encouraging that the pattern in the supermodels is nearly identical, showing that low frequency modes of variability are also synchronized and that connecting U, V, and T lead to the synchronization of the geopotential height field.

The principal components corresponding to the SUMO5 and SUMO6 PNA exhibit a Pearson correlation coefficient of 0.992, whereas the correlation between the CAM6 PNA PC and the SUMO6 PC is only 0.27. We would expect some correlation due
to tropical SST forcing in the AMIP runs (Wallace and Gutzler, 1981). Additionally, the PNA's variance explained sits well within the spread of the observations (Fig. 6e).

The NAO is slightly less synchronized (Fig. 3S) with a Pearson correlation coefficient of 0.75 between the two supermodel runs, but this correlation is still much higher than the correlation of -0.012 between the SUMO6 and CAM6 run.

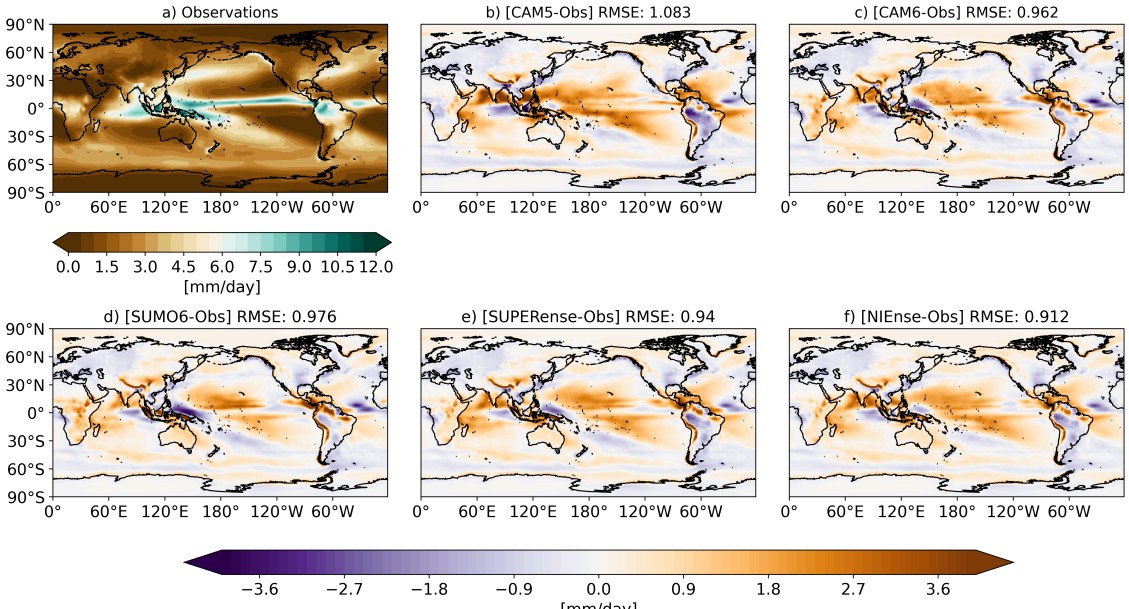

**Figure 7.** The observed annual precipitation climatology [mm/day] in the NOAA GPCP product (a) and the biases of the CAM5 model (b), CAM6 model (c), SUMO6 (d), SUMO5 (e), and NIEnse (f) relative to the observations. The model RMSE [mm/day] is shown in the title of each panel. The color bars indicate the precipitation bias in mm/day, with orange representing positive bias (too much precipitation) and purple representing negative biases (too little precipitation).

### 3.3 Impact on mean-field biases

One motivation for developing supermodels is their potential to reduce mean-field biases. In our work, we emphasize the supermodeling implementation connecting CESM components without performing any training—any bias improvements are muted. Therefore we do not anticipate substantial reduction in bias, but there may be minor improvements beyond that from averaging of non-interactive simulations because error compensation at an early stage (Schevenhoven et al., 2023; Duane and Shen, 2023). To ensure that the synchronization did not introduce significant errors or artifacts, we diagnose the climatological

biases. For most variables, the SUMO biases fall between those of CAM5 and CAM6 (see supplemental Table 1S for a statistics on the prognostic variables at multiple model levels), a pattern that holds true even when the fields are stratified by season (data not shown).

Figure 7 shows the annual precipitation climatology in the NOAA GPCP product and the model biases (Model - Observations). The SUMO5 and CAM5 precipitation biases are similar, likely because they share the same convection and boundary

layer schemes (see section 2.1.1 & 2.1.2). The same is true for the SUMO6 and CAM6 experiments.

In SUMO6 the largest differences from their respective constituent models are over the tropics with loading differences from the Bay of Bengal through the international dateline, and again off of the Pacific Coast of Central America (see Fig. 8a). This

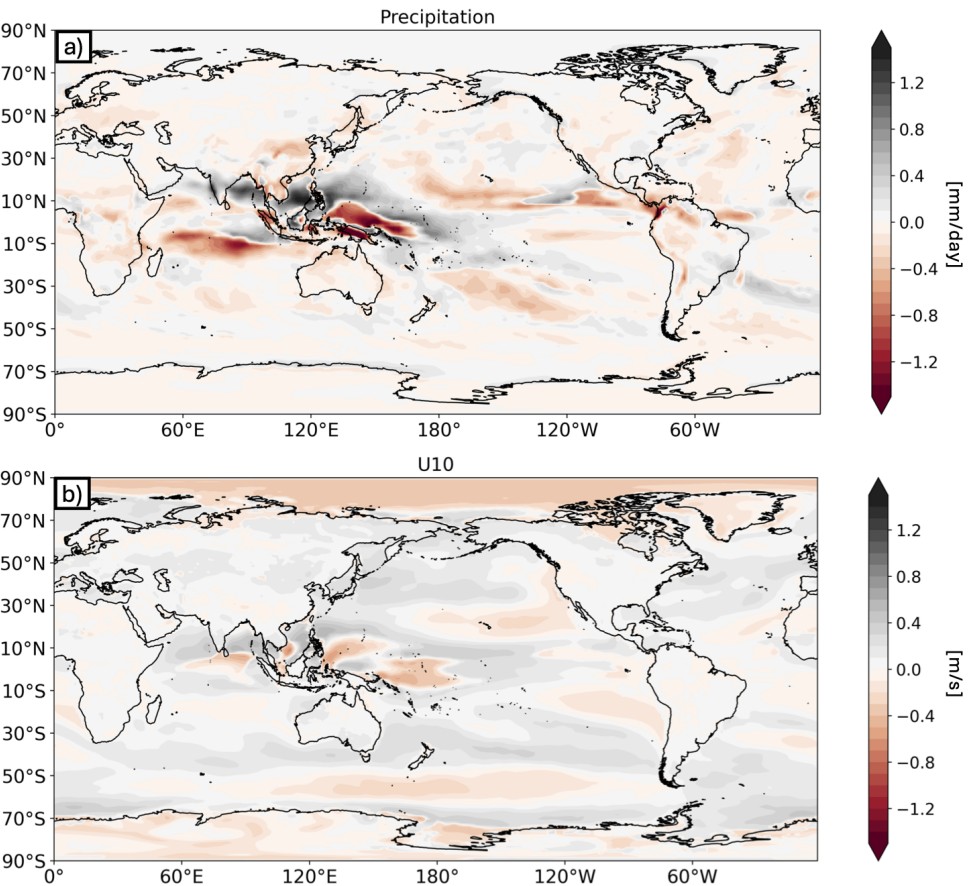

**Figure 8.** Annual climatological difference between SUMO6 and CAM6 for annual precipitation (a) and annual ten-meter winds (b).

indicates that the synchronization of the prognostic variables is likely affecting the monsoonal regions and deep convective zones.

We introduce a new experiment, the non-interactive ensemble (NIEnse), which is a multi-model ensemble mean of the CAM5 and CAM6 simulation runs, which interestingly has the lowest RMSE with a value of 0.91 mm/day. Figure 9 shows the same analysis but for the 10-meter wind speed. For wind speed, SUMO6 has an RMSE of 0.91 m/s, even outperforming the NIEnse simulation. The largest changes between the respective CAM and SUMO models occur over the maritime continent. This is particularly noticeable in the SUMO5 simulation (see Fig. 9e), which is closer to CAM6 than to CAM5 in this region.

To examine this more closely, we compare the differences between the NIEnse and an ensemble formed as the average of the two SUMO runs (SUPERense). Figure 10 shows the absolute difference in bias [(|NIEnse-Observations|) - (|SUPERense-Observations|)] for annually averaged precipitation (top) and U10 (bottom). Positive values (green) indicate that the SU-PERense is outperforming the NIEnse while negative values (blue) indicate the opposite. It is clear that the SUMOs formed

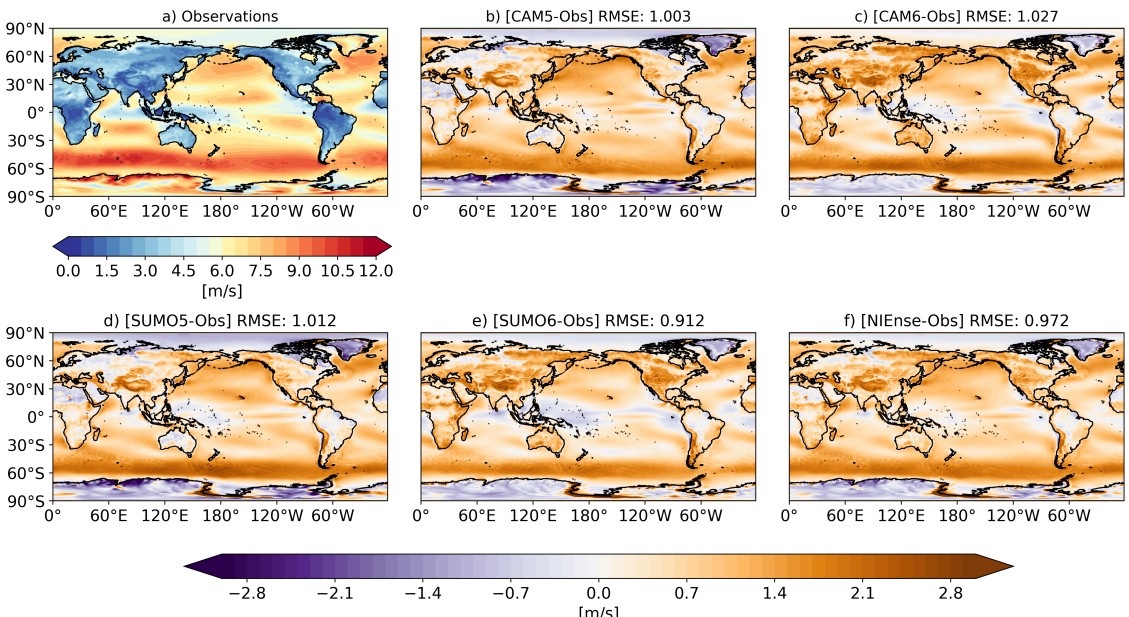

**Figure 9.** As in Figure 7, but for 10-meter wind speed.

their own dynamical systems with distinct biases. We observe that the SUPERense represents a 5% improvement to RMSE
over the NIEnse for annual U10 winds and a 3% degradation of annual precipitation when compared to the NIEnse.

Focusing on the U10 winds (Fig. 10b), the Pacific low-cloud deck regions are attenuated and degraded while the eastern boundary current regions are enhanced (see, for example, the Kurishio and Gulf stream extensions). Unlike, Counillon et al. (2023), we do not find that the areas of low synchronization necessarily lead to areas of high bias.

## 4 Conclusions

In this work, we give technical details on the implementation of the CAM5/CAM6 supermodel, which is the first to connect two atmospheric components of general circulation models in an HPC setting.

Our implementation leverages three new developments: 1) The exchange of information is managed through a novel Python-FORTRAN I/O interface that avoids the need to stop and start each model. This circumvents in CESM the costly initialization stage and introduces a PAUSE/RESUME capability (Fig. 1). This Python-FORTRAN bridge was also used to manage the
325 timestamps in the output files, and efficiently write the pseudo-observations files. 2) All component models are submitted through a single PBS or SLURM scheduler, which allows both component models to get into the same queue. This minimizes the time one component model has to wait for the other one to finish. Without these two improvements the supermodel would have been too slow to produce multi-year simulations. 3) We were able to adapt the CESM nudging toolbox (Davis et al., 2022) for our purposes, so that we have full control over e.g. which pseudo-observation variables we want to connect.

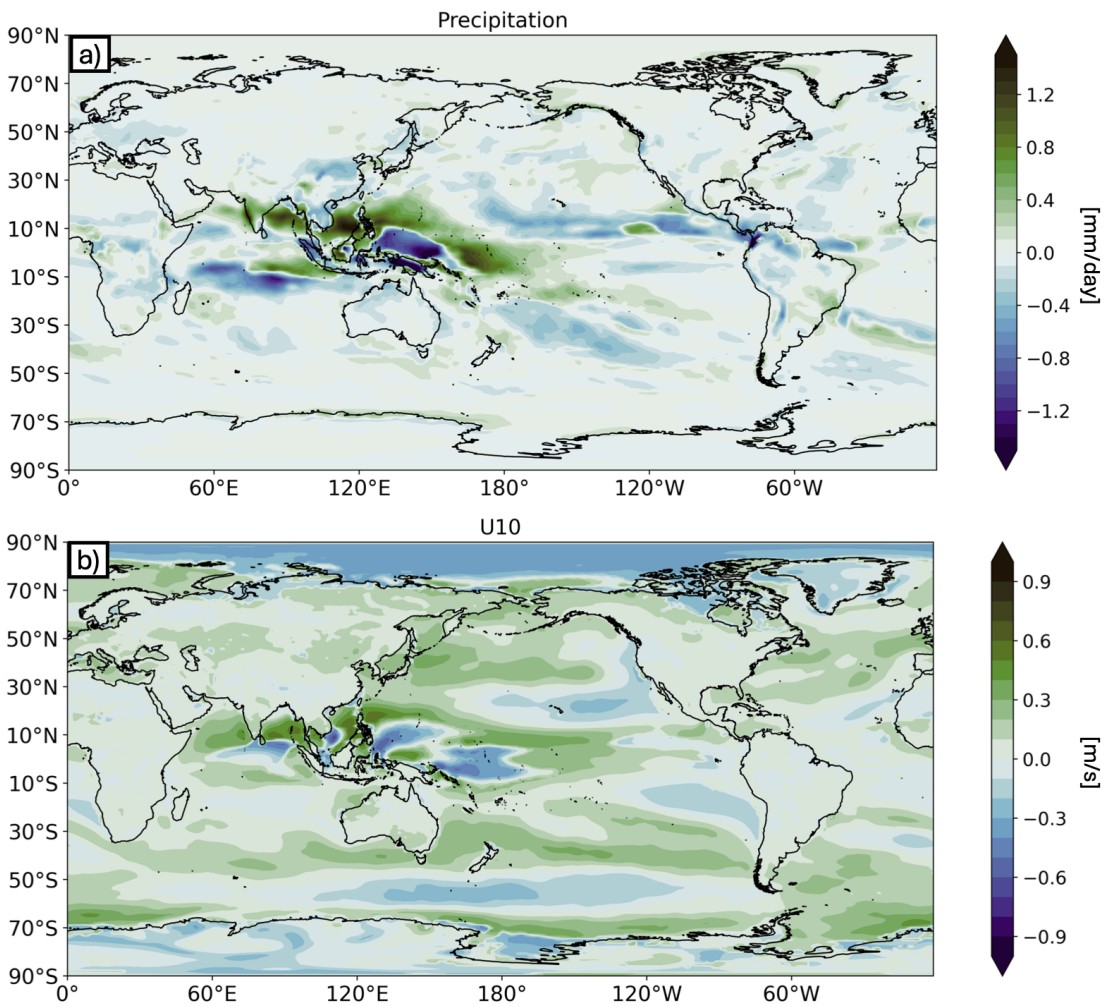

**Figure 10.** The annual absolute bias difference of NIense and the SUPERense ((|NIEnse-Observations|) - (|SUPERense-Observations|)) for precipitation (a) and 10-meter windspeed (b). Positive values indicate SUPERense is more skillful and vise-versa.

The supermodel framework is readily available for earth system research via our public GitHub repositories, making it relatively easy to port if active CESM systems are installed on a machine. Currently, the framework is available on both the NCAR and Norwegian supercomputers. Our driver scripts set-up the users constituent models and pseudo observations. All software is made available through Github repositories (see the code and data availability section).

To test our implementation we linked the CAM5/CAM6 atmosphere and confirmed that synchronization occurs across various temporal scales and variables. Additionally, even though the supermodels only exchange limited information (U, V, and T) every 6 hours, fields outside of the exchanged information exhibit synchronization across multiple time scales.

A key consideration for future work is assessing how the supermodel maintains physical consistency in terms of energy conservation. While our current analysis has focused primarily on wind and temperature fields, a thorough evaluation of radiative fluxes, surface turbulent fluxes, and the overall energy budget will be essential before extending this approach to coupled Earth system models. Ensuring an accurate energy balance will be crucial for improving the fidelity of the supermodel and avoiding unintended biases in simulations. A potential promising avenue of research would be to dynamically connect the fluxes in our supermodeling state as in Shen et al. (2017) which could ensure that each model's energy fluxes are accounted for in the supermodeling framework.

Additionally, our study has primarily examined large-scale variability modes such as the PNA and NAO. However, given the noted reduction in high-frequency variability over the tropics, it will be important to assess the impact on phenomena such as the Madden-Julian Oscillation (MJO) and convectively coupled equatorial waves. These modes play a key role in tropical variability and global teleconnections, and their representation within the supermodel framework remains an important avenue for future research.

In this study, we only nudge to pseudo-observations which are the equally-weighted mean of the two component models. Unweighted-mean supermodels can lead to partial synchronization regimes and localized variability damping (see also Counillon et al. (2023)). We find that in regions of lesser synchronization, some model variability is damped due to the smoothing effect of averaging over dissimilar fields, though the effects are minimal and should improve as the information exchange frequency ($\eta$) is increased.

Since our supermodel is untrained, we do not expect large improvements in the mean-fields biases. However, even with the untrained field, we find evidence of some improvement to the model climatological biases (see Fig. 9). We specifically examined localized structures for signs of improvement (Duane and Shen, 2023), but found no evidence to support any enhancement. With the computational efficiency of our implementation, we can now focus on the question if a trained supermodel of comprehensive Earth-System models can outperform its component models as demonstrated for simpler systems. Since our implementation is using the latest CESM infrastructure, an extension to coupled framework is straight forward.

Additional work will explore the use of machine learning techniques to dynamically optimize the weights and improve the performance of the CESM supermodel.

*Code and data availability.* To promote transparency and reproducibility, this study includes two code repositories:

1. All figure scripts are readily accessible and can be downloaded using the provided code on Zenodo (Chapman, 2025) to produce all figures.

2. To create all model runs and build your own supermodel, refer to Chapman et al. (2025). This second repository contains the setup for the SuperModel and its constituent models, including source modifications, model build scripts, and namelists for running the described CAM versions.

Comprehensive instructions for each step of this study are documented in the repository's README file. Raw ERA5 Reanalysis data can be obtained from the NSF NCAR Research Data Archive (European Centre for Medium-Range Weather Forecasts, 2019). The Global Precipitation Climatology Project (GPCP) Monthly Analysis Product data is provided by the NOAA PSL, Boulder, Colorado, USA, and can be accessed at https://psl.noaa.gov. CAM5 and CAM6, with directions to run, can be accessed at https://github.com/ESCOMP/CESM

*Author contributions.* WEC and FS developed the code to build, submit, and integrate the supermodels, led the output model diagnostics, and spearheaded the writing of the manuscript. JB and NK assisted in the interpretation of results, contributed to the writing, and secured project funding. IB, AKG, PC, and JN provided support in software engineering, contributed to the interpretation of results, and assisted in the writing of the manuscript.

*Competing interests.* The authors declare no competing interests

*Acknowledgements.* We would like to extend our gratitude to Jim Edwards for his invaluable guidance in software engineering for this project. We thank two anonymous reveiwers and the editor for their input which strengthened the manuscript. This work was supported by NSF project 2015618 - Coherent Precipitation Extremes in a Supermodel of Future Climate, ERC PoC grant number 101101037 and the Impetus4Change EU Horizon Europe project (grant no. 101081555). This research received support through Schmidt Sciences, LLC. We acknowledge discussions on synchronization and supermodeling with Dr. Greg S. Duane and Prof. Jeffrey B. Weiss. We gratefully acknowledge Norwegian HPC resources provided by sigma2 (NN9385K, NS9207k).

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
