# Peer review of "Implementation and validation of a supermodelling framework into CESM version 2.1.5"

_EGUsphere, 2024_

## Referee Comment (RC2)

Gescientific Model Development
Review: EGUsphere-2024-2682

Title: Implementation and validation of a supermodelling
framework into CESM version 2.1.5

Authors: Chapman, W. E. et al.
* * *
**Overview and major comments**

The present paper describes the implementation of a *supermodel* framework in which the two conventional climate models CAM5 and CAM6 are interacting, or *synchronized*, during their simulation through the regular exchange of nudging terms for some of their state variables. Through an appropriate tuning of the computation of these nudging terms, and because of the higher dimensionality of the supermodel benefiting from the advantages of each of its components, one might expect some compensation of the component model errors and an improved representation of the climate dynamical system. The present paper is a preliminary step towards such an assessment, providing a significant step in developing such kind of supermodels, sufficiently efficient to be used for climate studies.

The paper is well structured and written (though quite a few typos remain and deserve a more careful reading of the whole text). The objective are clearly stated, and the results clearly demonstrate an efficient supermodel (about 3-4 years of simulated years per days) and a rather appropriate synchronisation of the model variables, as indicated in particular by a high-frequency variability commensurate with the conventional models. I have a few general comments and a longer list of minor comments that follow. The general comments should not be understood as a major revision, as I consider the present paper as a technical contribution to the *Geoscientific Model Development* Journal. These comments are meant to widen a bit the analysis and whenever possible enhance the physical interpretation of the results and discuss their implications, possibly in light of previous works (which I am not familiar with).

**General comments**

1. The synchronisation is convincing for U, V and T, except over the tropics. Can you formulate hypotheses why this happens? Is it consistent with previous studies? To what extent is it an issue for the supermodelling strategy? Do you see ways to improve this synchronisation? For variables that are not part of the nudging strategy, the synchronisation is rather weak. To what extent is it also an issue for the supermodelling strategy?

2. Have you analysed the supermodel behaviour for other fields than wind and precipitation? What about radiative fluxes or surface turbulent fluxes? Do you keep a reasonable energy budget in the supermodels? If not, this should clearly prevent you to apply the approach for the coupled system, shouldn't it?

3. With respect to natural variability, you focus on the PNA and NAO types of variability. Have you analysed other modes of variability, like the MJO or convectively-coupled equatorial waves? To what extent is their simulated behaviour over the tropics consistent with the reduced high-frequency variability over the tropics?

**Minor comments**

1. p2, l28: the NMME and CMIP acronyms need to be defined.

2. p3, l60: typo: one of the two 'to be' needs to be removed.

3. p3, l80: what does 'reference' stand for here? Did you forget to add a reference here?

4. p3, l81: I guess it is 'component models' rather than 'components model'.

5. p4, l123-124: do you mean that sea surface temperatures are constant over each day in CAM (there is thus a small jump at the end of each day)?

6. p5, l127: why using a bilinear interpolation and not a conservative one? At least for precipitation, a conservative interpolation sounds more appropriate. Besides, what is the resolution of ERA5 and GPCP datasets?

7. p5, l151-154: I feel this technical development requires a bit more explanation to more fully understand how you overcome this challenge of submitting jobs through a single PBS/SLURM scheduler.

8. Section 2.4: I am a bit confused about how the nudging is performed. Do you average the instantaneous state of the atmosphere over the two model, and then use it for nudging over the 6 following hours (thus the

fields toward which the model is nudged are constant over the 6-hour window)? Besides, which nudging timescale to you use?

9. p6, l167-168: while being an important effort toward open science, this sentence does not seem to be at the right place in these technical description.

10. p7, l171-172: this would be interesting to have the elapsed (integrated also) time also for CAM5 and CAM6, to document the overloading of the model synchronisation.

11. p7, l177: can you elaborate a bit more on this difficulty when adding specific humidity in the nudged state variables?

12. p8, l195: I guess your refer to Figure 3.

13. p8, l196: 'Fig.' is missing before '3'.

14. p8, l199-200: without any more detailed analysis, I would argue that the whole atmospheric physics might be at play (most of it is strongly different between CAM5 and CAM6). Besides because the U, V and T forcing in the tropics is in general weaker than in the extratropics, this is rather expected that the model are more sensitive to their own physics, isn't it?

15. p8, l202: missing ending bracket.

16. p8, l202-203: The link between the two parts of the sentence remains unclear, and not obviously consistent with what you write l199-200.

17. p8, l207-208: the link is interesting, but probably hard to fully understand for most readers. A bit more explanation would be welcome.

18. p8, l218: missing closing bracket.

19. p10, l222: do you mean increasing or reducing the relaxation timescale?

20. p12, l251: do you mean the correlation between the PNA pattern of SUMO5 and that of SUMO6?

21. p12, l255: do you have an interpretation or an hypothesis for such a different result between PNA and NAO? Has it been seen in previous studies?

22. p13, l258: I guess it should be 'here', not 'her'.

23. p13, l267-268: My understanding is that Figure 4S is showing the differences between the non-interacting ensemble and the supermodel ensemble. It does not seem to correspond to what you are referring to here.

24. p16, l289: I would remove the 'in' before CESM.

25. p17, Code and data availability: a recap about the CAM versions and the place where to find the code would be welcome in this section.

---

## Editor Decision (ED1)

Dear Author,

Thank you for your revised manuscript.

I have to say that I find it somewhat disturbing that you submitted a first manuscript with so many typos. This is, for me, not very respectful for the reviewers that then have to take the time to list them. Fortunately, the form has been improved thanks to the numerous corrections you included in the revised manuscript following the reviewers' remarks.

It also seems that you have answered most of the more fundamental reviewers' comment on the content. However, it is hard for me to make a firm judgement on this as you did not, in your reply attached with your revised manuscript, always follow the rule that is to provide a clear identification of your changes in the manuscript for each of the reviewers' remarks (see https://www.geoscientific-model-development.net/submission.html#articlefiles):

> •The **author's response** in case of revisions must be submitted as one separate *.pdf file (indicating page and line numbers), structured in a clear and easy-to-follow sequence: (1) comments from referees/public, (2) author's response, and (3) author's changes in the manuscript.

Indeed, you followed this rule for your answers to the minor comments but not for the (most important) general comments. Therefore, I would like you to review your reply egusphere-2024-2682-author_response-version2.pdf , **indicating precisely what you changed in the text** in relation with your answers :

- p.2 : « We now explicitly link to a separate repository that reconstructs every figure from the manuscript. This was included in the original submission but may not have been sufficiently highlighted. It is now directly referenced both in the manuscript and the README. »

- p.2 : « We have added further instructions on obtaining and installing CESM2.1.5 and the associated CONDA python environments to ensure users can set up the required environment, including explicit instructions on necessary modifications. »

- p.2 : « Added explicit instructions for users to install their own version of CESM2.1.5. »

- p.2 : « Clearly indicated where modifications should be made within the codebase to adapt it to different computing environments. »

- p.3 : « Include a more detailed comparison of improved and degraded fields of the prognostic variables (U,V,T,Q,PS) at multiple model levels. »

- p.3 : «Included an RMSE table in the supplemental to capture the model state biases for the reader. »

- p.3 : «Improved upon our insights into why certain fields show improvement while others do not. »

- p.3 : «Clarify that while tuning will play a role in further optimizing performance, some improvements can already be observed with equal weighting. »

- p.4 : «Generally, in the manuscript, we have de-emphasized the point that the model could contain lower biases and instead focus on the platform specifications and the system. »

- p.6 : The paragraph on the «Synchronization in the Tropics and for Non-Nudged Variables », especially as you wrote « Future work should explore whether increased nudging frequency or selective tuning of nudging coefficients could help address this issue »

- p.7 : The paragraph on the «Energy Budget and Additional Fields »

- p.7 : The paragraph on the « Modes of Variability in the Tropics »

I also have the additional following comments that I would like you to take into account :

- On p.3 of you reply, you wrote « Duane and Shen (2023) reveal that often improvements are manifest in representations of localized structures, rather than in reductions in RMS error. We have searched for evidence of this, but find no direct measure of improvement in our system. » : can you add something in your paper to clarify that you did not find any improvement in the representations of localized structures in your system ?

- On p.4 of your reply, you wrote « There are no computational or physical constraints, though the models have been tuned at NCAR to represent fields well in their creation, likely this is why the distributions are similar. » ; can you add something in your paper along those lines ?

- On p. 8 of your reply, there is an insert «... rather than decades. Though, decadal non-local corrections have been made in a simplified coupled ocean-atmosphere model (Brajard et al. 2021). » ; what is this about ? I don't find this is the manuscript.

- On p.7 of your updated manuscript, the comma should be replaced by a full stop in « … period 1979 through 2005, The supermodel …. »

- On p.14 of your updated manuscript, you refer to Fig. 4S. If you refer to it, it should be included in the paper (not only in the supplementary material).

- On p.2 of your manuscript, I think you should mention up front that no training has been used in your current supermodel. On p.4, you mention « We also use the above-described training methods to optimize the performance of the supermodel, the results of which will be described elsewhere. » ; please be more specific about the « elsewhere » !

- On p. 5 of your manuscript, please revise the part of the sentence starting with as well as in « … which are linearly interpolated to obtain specified independent values at each time-step, as well as the evolution of aerosol emissions and trace gas concentrations (including $CO_2$). » ; the current sentence reads awkward to me

- Finally, I am surprised that you implement your PAUSE/RESUME capability instead of using a dedicated coupling software such as OASIS (https://oasis.cerfacs.fr/), MCT (https://web.cels.anl.gov/projects/climate/mct/) or YAC (https://dkrz-sw.gitlab-pages.dkrz.de/yac/index.html) . Can you add any justification on this ?

With best regards, looking for your reply,

Sophie

---

## Author Response (AR2)

**Authors' Response to Reviews of**

**Implementation and validation of a supermodelling framework into CESM version 2.1.5**

W.E. Chapman et al.
*Geoscientific Model Development*, `EGUsphere-2024-2682`
* * *
**EC:** *Editor's Comment*, **RC:** *Reviewers' Comment*,     AR: Authors' Response,     ☐ Manuscript Text

We thank the reviewers and editor for their thoughtful comments, and have included an acknowledgment to their work in the manuscript.

**EC:** *Thank you for your revised manuscript.*

**EC:** *I have to say that I find it somewhat disturbing that you submitted a first manuscript with so many typos. This is, for me, not very respectful for the reviewers that then have to take the time to list them. Fortunately, the form has been improved thanks to the numerous corrections you included in the revised manuscript following the reviewers' remarks.*

**EC:** *It also seems that you have answered most of the more fundamental reviewers' comments on the content. However, it is hard for me to make a firm judgment on this as you did not, in the reply attached with your revised manuscript, always follow the rule that is to provide a clear identification of your changes in the manuscript for each of the reviewers' remarks. I have listed the points for which I would like to have this clearer identification and I have made few additional comments in the file here attached.*

**EC:** *Thanks in advance for your replies to my comments before I can consider a potential publication of the manuscript.*

**0.1. Author response:**

We thank the editor for their time and thoughtful effort in improving our manuscript. We appreciate the detailed feedback provided, and have included an acknowledgment to their work in the manuscript.

We regret that the initial submission contained more typos than appropriate, and we fully acknowledge that this may have caused additional work for the reviewers. We apologize for this oversight and appreciate the corrections and suggestions that helped us improve the manuscript. These issues have now been carefully addressed.

Regarding the identification of changes: as this is the authors' first open-access article, we were unfamiliar with some of the norms specific to this review format. In the open-access system, the revisions are not typically posted with line-by-line tracked changes, and we had assumed that line numbers in the response letter would not be particularly useful in this context. However, we appreciate the editor's request for clearer identification of the changes, and we have now revised our response accordingly to explicitly indicate where changes have been made in the manuscript, point-by-point to the editor's requests. Thank you for forwarding the hyper-link to the revision standards.

With respect to Reviewer 1's major concerns, we note that many of their comments focused on the code implementation and documentation. While several of these issues were related to the code base itself rather than the text of the manuscript, we have amended the documentation and clarified these points in both

the manuscript and the accompanying materials. We point to the code base where appropriate, and to the manuscript where appropriate. We hope these changes now meet the expectations of the editor and reviewers. Below we have included the line references to the updated changes as the editor has requested in their review document to each comment. We note that Reviewer 2 suggested general changes that "*should not be understood as a major revision, as I (they) consider the present paper as a technical contribution to the Geoscientific Model Development Journal*", and address their comments as such.

Finally, we wish to emphasize our respect for the time and expertise of all reviewers and editors. We remain committed to engaging in this process in a collegial and professional manner and hope that the dialogue around this manuscript can continue in that spirit.

We thank the editor again for their careful reading and valuable feedback and look forward to their further consideration of the manuscript.

Below you will find the line-by-line corrections requested by the Editor included in the appropriate sections of the reviewer comments. As this is a new review, we have created our new track changes from this review, but refer the editor to the previous track changes for references to the reviewer comments below.

**EC:** *Indeed, you followed this rule for your answers to the minor comments but not for the (most important) general comments. Therefore, I would like you to review your reply egusphere-2024-2682-author-response-version2.pdf , indicating precisely what you changed in the text in relation with your answers :*

AR: For every point raised by the reviewer, we have added the associated line numbers in our response to the authors below. For convenience of the reviewer, we repeat the review document with the added line numbers.

**1. Editor Comments:**

**EC:** *On p.3 of you reply, you wrote « Duane and Shen (2023) reveal that often improvements aremanifest in representations of localized structures, rather than in reductions in RMS error. We have searched for evidence of this, but find no direct measure of improvement in our system. » : can you add something in your paper to clarify that you did not find any improvement in the representations of localized structures in your system?*

AR: We have added the language to L350:

> Since our supermodel is untrained, we do not expect big improvements in the mean-fields biases. We specifically examined localized structures for signs of improvement [Duane and Shen, 2023], but found no evidence to support any enhancement.

**EC:** *On p.4 of your reply, you wrote « There are no computational or physical constraints, though the models have been tuned at NCAR to represent fields well in their creation, likely this is why the distributions are similar. » ; can you add something in your paper along those lines?*

AR: We have added that language to L240:

> To examine the supermodel for signs of significant variance deflation, we compare histograms of 6-hourly averaged wind speed values (Fig. 4). We note that the CAM5 (red-dashed) and CAM6 (black-dashed) distributions are quite similar, this is likely do to the model tuning activity at NCAR

> prior to the model release. We detect a slight damping of the background winds near the mode of the distribution, but no degradation of the highest wind speeds. Overall, the difference between the component and super-models is minimal.

**EC:** *On p. 8 of your reply, there is an insert «... rather than decades. Though, decadal non-local corrections have been made in a simplified coupled ocean-atmosphere model (Brajard et al. 2021). » ; what is this about ? I don't find this is the manuscript.*

AR: This was a proposed revision to a sentence that has been removed from the manuscript. We apologize for any confusion it has caused.

**EC:** *On p.7 of your updated manuscript, the comma should be replaced by a full stop in « ...  period 1979 through 2005, The supermodel .... »*

AR: Thank you, this is ammended.

**EC:** *On p.14 of your updated manuscript, you refer to Fig. 4S. If you refer to it, it should be included in the paper (not only in the supplementary material).*

AR: Done. Please see L300:

> In SUMO6 the largest differences from their respective constituent models are over the tropics with loading differences from the Bay of Bengal through the international dateline, and again off of the Pacific Coast of Central America (see Fig.  8a). This indicates that the synchronization of the prognostic variables is likely affecting the monsoonal regions and deep convective zones.

**EC:** *On p.2 of your manuscript, I think you should mention up front that no training has been used in your current supermodel. On p.4, you mention « We also use the above-described training methods to optimize the performance of the supermodel, the results of which will be described elsewhere. » ; please be more specific about the « elsewhere » !*

AR: We note that the model is untrained in the Abstract, and have added an aditional sentence to ensure the reader is aware of the results. We also, ammend the word "elsewhere".

> **Abstract:** In this study, we examine a single untrained supermodel where each model version is equally weighted in creating pseudo-observations.

Line 63:

> To achieve optimal performance, a supermodel must be trained using data from the "truth" such as observations or a reference model. During the training phase, the supermodel interaction coefficients are optimized to formulate the supermodel with the best skill. Since only the interaction coefficients need to be learned, the training effort is substantially less than that used by modern machine learning approaches which learn the entire forward operator of a model [e.g., Watt-Meyer et al., 2023]. Efficient methods have been developed to train supermodels [Schevenhoven and Selten, 2017] and have been shown in a coupled model of intermediate complexity, SPEEDO, connected via the atmospheres only [Severijns and Hazeleger, 2010], these training methods showed promising results [Schevenhoven et al., 2019], even when the observations were sparse and noisy [Schevenhoven, 2021]. As the goal of this

> manuscript is to describe the modeling framework itself, we will henceforth focus on results from an untrained supermodel.

Line 95

> We also use the above-described training methods to optimize the performance of the supermodel, the results of which will be described  in a forthcoming manuscript.

**EC:** *On p. 5 of your manuscript, please revise the part of the sentence starting with as well as in« … which are linearly interpolated to obtain specified independent values at each time-step, as well as the evolution of aerosol emissions and trace gas concentrations (including CO2). » ; the current sentence reads awkward to me*

AR: This sentence has been amended:

>
>
> The model simulations followed the protocol of the Atmospheric Model Intercomparison Project (AMIP) and are forced by observed monthly sea surface temperatures and sea ice from 1979 to 2005 (26 years), with values linearly interpolated at each time step. The simulations also include prescribed evolutions of aerosol emissions and trace gas concentrations (including $CO_2$).

**EC:** *Finally, I am surprised that you implement your PAUSE/RESUME capability instead of using a dedicated coupling software such as OASIS (https://oasis.cerfacs.fr/), MCT (https://web.cels.anl.gov/projects/climate/mct/) or YAC (https://dkrz-sw.gitlabpages.dkrz.de/yac/index.html). Can you add any justification on this ?*

AR: Coupling software such as OASIS or YAC have been critical to the evolution of modern climate modeling. They typically would be very logical choices for implementing supermodeling techniques. However, there are some limitations in the context of this framework that made them less than ideal for our task and PAUSE/RESUME was more appropriate:

1. The MCT coupler of CESM does not support exchange of 3d variables which is critical for this work.

2. CESM supports having multiple instances of a component model; however, all these instances must use the exact same code, which is not the case in our study.

3. The pause/resume approach with disk-based communication can with little effort be ported to ESMs that feature different couplers (e.g., CAM7 will use NUOPCY/ESMF instead of MCT

However, we wanted to acknowledge this point by the editor, we have added the following text to the manuscript at Line 160:

> If the component model grids differ, Python interpolation routines are invoked to ensure consistency. Once the output has been processed, the Python script removes the `PAUSE` file, allowing the model to resume operation without the need for re-initialization or re-entering the queue. The implementation of efficient mpi-based communication between the models (i.e. standard coupling software) was beyond

the scope of the study but is something that should be explored in future efforts.

**2. Reviewer #1**

**RC:** *This paper describes a supermodelling framework that combines the effects of the physics parameterization suites from two versions of the Community Atmosphere Model. The physics suites are combined by nudging each version (model component) to the averaged state on a periodic time interval, as in data assimilation.*

**RC:** *Supermodelling is an interesting idea with interesting potential applications, but I think that the manuscript is not ready for publication for three reasons. First, the method is not described in enough detail to allow a reader to reproduce the results. For instance, I couldn't find a tag of the code listed in the manuscript that would allow me to recreate the figures in the manuscript. Furthermore, the scripts have hardwired paths (not variables) with little instruction given as to what lines in the scripts need to be modified. Second, the main application cited is improvement of supermodeled climatologies over the individual components, but only one or two improved fields are shown in the manuscript. The authors argue that improvement will come with tuning, but one might expect to see improvement even with simple equal weighting of the components. It might interest readers to see more discussion of which fields are improved and which are degraded, along with any insights that the authors have regarding the reasons that some fields are improved and others are not. Third, there are many typos in the manuscript, some of which impede understanding of the meaning. I list just a sample of them below. My recommendation is that the authors invest more time in the manuscript and deliver a more polished version.*

**3. Reviewer #1**

**3.1.** **Author response:**

We would like to sincerely thank the reviewer for their time and effort in evaluating our manuscript. We appreciate the constructive points raised and have made substantial revisions to improve clarity, documentation, and presentation of results. Below, we address each of the major concerns raised by the reviewer.

**1. Clarity and Readability of the Manuscript**

We acknowledge the reviewer's comment regarding typos and overall polish. In response, we have carefully revised the manuscript to improve clarity, correct errors, and enhance readability. We appreciate this feedback and believe that the current version is significantly more refined.

**2. Reproducibility and Code Documentation**

We understand the importance of reproducibility and have taken significant steps to improve documentation. The original manuscript included a GitHub repository containing all necessary code to reproduce the figures, but we recognize that the accompanying documentation may not have been sufficiently clear. We highlight that this is not the same repository that the supermodel built is in, which is stated in the **Code and data availability** section. This link to the repository has been highlighted in the supermodel REPO. Additionally, we have expanded on how a user can port their supermodel on any new machine. To address this:

- We have substantially expanded the README and associated documentation to provide explicit guidance on setting up and running the supermodel framework. **See Line 195, Line 334, and 368. Additionally, see the dramatically improved github README [GITHUB LINK] which is reflected in the static ZENODO release.**

- We now explicitly link to a separate repository that reconstructs every figure from the manuscript. This was included in the original submission but may not have been sufficiently highlighted. It is now directly referenced both in the manuscript and the README. **See Line 366.**

- We have added further instructions on obtaining and installing CESM2.1.5 and the associated CONDA python environments to ensure users can set up the required environment, including explicit instructions on necessary modifications. **See the github README [GITHUB LINK].**

We believe these changes significantly enhance the reproducibility of our work.

**3. Model Configuration and Hardcoded Paths**

We would like to clarify that our implementation did not contain hardcoded paths except for those referencing the supported instance of CESM2.1.5. However, we recognize that clearer guidance on configuring paths for other environments would be beneficial. To this end, we have:

- Added explicit instructions for users to install their own version of CESM2.1.5. **See the github README [GITHUB LINK].**

- Clearly indicated where modifications should be made within the codebase to adapt it to different computing environments. **See the github README [GITHUB LINK].**

**4. Evaluation of Model Performance**

The reviewer notes that the manuscript primarily focuses on introducing the platform and does not extensively analyze the improvement of supermodeled climatologies. While we maintain that the primary goal of this work is to introduce the framework, we agree that additional analysis can provide valuable context. Additionally, we have removed the sentence in the abstract that may mislead readers and now it is focused more on the platform itself. While these account for extensive changes at multiple points in the manuscript we have tried to capture and highlight the location of the changes below. Generally, we encourage the reviewer to see the "mean-field biases" section of the manuscript (Section 3.3). We have expanded on the following points:

- Include a more detailed comparison of improved and degraded fields of the prognostic variables (U,V,T,Q,PS) at multiple model levels. **See revised section 3.3 "Impact on mean-field biases" in the track changes document as well as the supplemental table which shows the RMSE of the multiple model state variables at multiple levels.**

- Included an RMSE table in the supplemental to capture the model state biases for the reader. **See Line 299 and the supplemental material.**

- Improved upon our insights into why certain fields show improvement while others do not. **For examples of this see lines 205-210, 234, 257**

- Clarify that while tuning will play a role in further optimizing performance, some improvements can already be observed with equal weighting. **See section 3.3 Impact on mean-field biases and lines 358-362 as well as the discussion of localized structures in the remaining part of this response.**

These additions should help readers better understand both the potential and limitations of the approach.

**Final Note**

We appreciate the reviewer's feedback and believe that the revisions significantly enhance the manuscript. We hope that the changes we have made address all concerns and improve the clarity and reproducibility of our work. Thank you again for your time and consideration.

**3.2. Minor Comments**

RC: *This part of the caption of Fig. 2 seems to have typos: "SUMO5 (teal); supermodel which uses CAM5-physics (SUMO5); supermodel supermodel which uses CAM6-physics (SUMO6) (purple) at location"*

> ...  ... ... a supermodel which uses CAM5-physics (SUMO5; teal); a supermodel which uses CAM6-physics (SUMO6; purple)...

**RC:** *Lines 32–33: "Additionally, biases which are shared across the individual models in an NIE cannot be corrected due to the linear nature of post-process averaging." But does the supermodel succeed in correcting such shared biases? Can the authors offer any insight into when and why shared biases can be corrected?*

**AR:** This line has been adjusted to point to literature in which the improvement of shared biases is indeed improved in supermodeling. We refer the reader to the revised text. We also encourage the reviewer to read the community comment by Gregory Duane, an expert in super modeling. Duane and Shen (2023) reveal that often improvements are manifest in representations of localized structures, rather than in reductions in RMS error. We have searched for evidence of this, but find no direct measure of improvement in our system. We have adjusted the text to include these references. Please see lines 37-40 in the new manuscript.

**RC:** *Line 80: "the adaptation of the existing nudging toolbox (reference)". Typo. Please include whatever reference you're referring to here.*

**AR:** Fixed, Thank you.

**RC:** *Lines 178–180: "We show results for four experiments: CAM5, CAM6, the supermodel which uses CAM5-physics, but is nudged to the combined state, SUMO5, and the supermodel which uses CAM6-physics, SUMO6." Does SUMO6 nudge to the combined state? If not, how is it different from CAM6? The sentence ends before the SUMO6 configuration is clearly described.*

**AR:** We apologize for the confusion, we were using shorthand though it is best to be explicit. SUMO6 is also nudged to the combined state. This has been fixed in the text.

> We show results for four experiments: CAM5, CAM6, the supermodel which uses CAM5-physics, but is nudged to the combined state, SUMO5, and the supermodel which uses CAM6-physics, but is nudged to the combined state, SUMO6.

**RC:** *Line 195: "(Fig. ??, right column)". Typo.*

**AR:** This has been fixed, thank you.

**RC:** *Lines 195–196: "Correlations between the SUMO5 and SUMO6 experiments are much higher (3, left column)" How can a correlation be as high as 3? Correlations must lie between -1 and 1.*

**AR:** The 3 refers to pointing the reader at Figure 3, not a correlation score. We have made this more clear in the text.

**RC:** *Fig. 4: I'm surprised that CAM5 and CAM6 have such similar distributions of wind speed. Is there a computational or physical constraint on this distribution?*

**AR:** There are no computational or physical constraints, though the models have been tuned at NCAR to represent fields well in their creation, likely this is why the distributions are similar.

**RC:** *Line 258: "One reason to develop supermodels lies in their potential to have smaller mean-field biases." In the example of surface precipitation shown in Fig. 7, the spatial pattern of errors looks similar (shared) in CAM5 and CAM6. The hope implied in lines 32–33 was that such error could be reduced below both components by supermodelling. But Fig. 7 shows that this doesn't happen for surface precipitation. Do the authors have an explanation for this failure?*

**AR:** Currently, our theory is that when we train the super model, these biases will all improve. However, for now

we point to the under synchronization in the tropics as a potential cause. Generally, in the manuscript, we have de-emphasized the point that the model could contain lower biases and instead focus on the platform specifications and the system. While this is present in several locations, we highlight a few below:

For example, in the Abstract we now say (Line5):

> Here we present a research framework for the first atmosphere-connected supermodel using state-of-the-art atmospheric models. The Community Atmosphere Model (CAM) versions 5 and 6 exchange information interactively while running, a process known as supermodeling. The primary goal of this approach is to synchronize the models, allowing them to  create a new dynamical system which can benefit from each component model, in part by increasing the  dimensionality of the system.

We encourage the reviewer to see the first paragraph of the rewritten **section 3.3** which demonstrates this de-emphasis of the bias improvements most directly.

**RC:** *Line 258: "The work presented her" Typo.*

 AR: Fixed! Thank you.

**RC:** *Lines 278–279: "Positive values (grey) indicate that the SUPERense is outperforming the NIEnse and vise-versa." Are positive values grey or green?*

> Positive values (green ) indicate that the SUPERense is outperforming the NIEnse  while negative values (blue) indicate the opposite.

**RC:** *Lines 289–290: "This circumvents the in CESM costly initialization stage and introduces effectively PAUSE/RESUME capability (Fig. 1)." Typos?*

 AR: Yes, we have fixed the typos.

> The exchange of information is managed through a novel Python-FORTRAN I/O interface that avoids the need to stop and start each model.  This circumvents in CESM the costly initialization stage and introduces a PAUSE/RESUME capability (Fig. 1). This Python-FORTRAN bridge was also used to manage the timestamps in the output files, and efficiently write the pseudo-observations files.

**RC:** *Lines 300-301: "To test our implementation we linked the CAM5/CAM6 atmosphere and confirmed that synchronization across various temporal scales and variables Even though the supermodels only exchange limited information . . ." What did you confirm?*

 AR: We have fixed the typo that clarifies the text.

> To test our implementation we linked the CAM5/CAM6 atmosphere and confirmed that synchronization occurs across various temporal scales and variables.

**RC:** *Lines 317–319: "To create all model runs and build your own supermodel, refer to Chapman et al. (2024). This second repository contains the setup for the SuperModel and its constituent models, including source*

*modifications, model build scripts, and namelists for running the described CAM versions." Is a tag of the CESM repository listed somewhere so that readers can reproduce the figures in the paper identically?*

AR:  Yes, though this was in the original manuscript, we have highlighted the text in the model README so it is easier to find and link to the figure repository in the new documentation.

We thank the reviewers for their thoughtful comments, and have included an acknowledgment to their work in the manuscript. We highlight that the manuscript has gone through a few significant changes due to this review and we believe it is in a much stronger condition. It has been refined greatly with an eye towards readability and flow. We include many more error statistics (see table 1 in the supplemental material). The github repository has been restructured for user clarity (which was a concern of reviewer 1, and this has resulted in new versions on zenodo), and we highlight the improved use-ability of the code and modeling framework. Additionally, the GPCP dataset has been remapped with a new interpolation and this updated many figures, though the broad findings are unchanged.

Though you had only included questions as minor points to strengthen the manuscript, we have incorporated those points into our revised manuscript and share the main changes of those points below.

**4.  Reviewer #2**

RC:  *The present paper describes the implementation of a supermodel framework in which the two conventional climate models CAM5 and CAM6 are interacting, or synchronized, during their simulation through the regular exchange of nudging terms for some of their state variables. Through an appropriate tuning of the computation of these nudging terms, and because of the higher dimensionality of the supermodel benefiting from the advantages of each of its components, one might expect some compensation of the component model errors and an improved representation of the climate dynamical system. The present paper is a preliminary step towards such an assessment, providing a significant step in developing such kind of supermodels, sufficiently efficient to be used for climate studies. The paper is well structured and written (though quite a few typos remain and deserve a more careful reading of the whole text). The objective are clearly stated, and the results clearly demonstrate an efficient supermodel (about 3-4 years of simulated years per days) and a rather appropriate synchronisation of the model variables, as indicated in particular by a high frequency variability commensurate with the conventional models. I have a few general comments and a longer list of minor comments that follow. The general comments should not be understood as a major revision, as I consider the present paper as a technical contribution to the Geoscientific Model Development Journal. These comments are meant to widen a bit the analysis and whenever possible enhance the physical interpretation of the results and discuss their implications, possibly in light of previous works (which I am not familiar with).*

**4.1.  General Comments**

RC:  *1. The synchronisation is convincing for U, V and T, except over the tropics. Can you formulate hypotheses why this happens? Is it consistent with previous studies? To what extent is it an issue for the supermodelling strategy? Do you see ways to improve this synchronisation? For variables that are not part of the nudging strategy, the synchronisation is rather weak. To what extent is it also an issue for the supermodelling strategy?*

RC:  *2. Have you analysed the supermodel behavior for other fields than wind and precipitation? What about radiative fluxes or surface turbulent fluxes? Do you keep a reasonable energy budget in the supermodels?*

*If not, this should clearly prevent you to apply the approach for the coupled system, shouldn't it?*

**RC:** *3. With respect to natural variability, you focus on the PNA and NAO types of variability. Have you analysed other modes of variability, like the MJO or convectively-coupled equatorial waves? To what extent is their simulated behaviour over the tropics consistent with the reduced high-frequency variability over the tropics?*

We sincerely thank the reviewer for their insightful feedback, which raises important questions about the synchronization characteristics and overall performance of the supermodel. Below, we address each point in turn and outline directions for further investigation. We have altered the text in the manuscript to address all of the points, or to point to new avenues of research, in the manuscript.

**1. Synchronization in the Tropics and for Non-Nudged Variables**

We acknowledge the reviewer's observation that synchronization is weaker in the tropics and for variables outside the nudging strategy. One possible explanation for this is that tropical dynamics are more governed by convective and diabatic processes rather than large-scale balanced dynamics, making them inherently more chaotic and less constrained by the imposed nudging. Previous studies (e.g., [Counillon et al., 2023] ) have also observed similar challenges in achieving full synchronization in the tropics, suggesting this may be a fundamental limitation of the approach when applied to regions dominated by deep convection. However, there is hope for improvements in coupled models because surface fluxes can be synchronized via sea surface temperature leading to improvements in climatological precipitation patterns in the tropics, as seen in Shen et al. [2016], in the COSMOS model. This is because of the ocean-atmosphere interaction is strongest in the tropics.

For variables that are not explicitly nudged, weaker synchronization is expected, as their evolution remains primarily determined by the underlying physical parameterizations of each component model. While this does not necessarily undermine the supermodelling approach, it does suggest that expanding the set of exchanged variables or optimizing the nudging parameters could improve synchronization. Future work should explore whether increased nudging frequency or selective tuning of nudging coefficients could help address this issue.

The changes made to the manuscript are in multiple places here. We refer the reviewer to the **third and seventh paragraphs in Section 3.1 and the first paragraph in section 3.3** to see a demonstration of those changes;

As well as Line 205:

> In this implementation, the information in U, V, and T are exchanged and nudged while Q is left to evolve freely. We speculate that the main challenges with including specific humidity (Q) in the nudging process stem from the intrinsic properties of moisture in the atmosphere and its coupling with cloud and precipitation processes. This was done because previous work indicated difficulty when adjusting specific humidity Q in CAM in both, nudging (e.g., Chapman and Berner, 2024) and full DA experiments (Raeder et al., 2021).

and Line 257:

> There is a significant damping of variability in this frequency band in areas where we find lower model synchronization like the tropics and the poles (Fig. 3a & c), especially over the maritime

continent.  In nudging studies, moving to an observations frequency of less than 6 hours seems to alleviate effects of damping (Davis et al., 2022), so we hypothesize that increasing the SUMO interaction frequency will be beneficial with regard to minimizing the damping of high frequency variability.

**2. Energy Budget and Additional Fields**

The reviewer raises an important point regarding radiative and surface turbulent fluxes, as well as the overall energy budget of the supermodel. While our current analysis has primarily focused on wind and temperature fields, we recognize that maintaining a reasonable energy balance is crucial, particularly for future applications in coupled models.

Although we have not conducted a detailed analysis of radiative fluxes or surface energy budgets in this study, this is a key direction for further verification. Ensuring that the supermodel maintains physical consistency in terms of energy conservation will be necessary before extending the approach to coupled systems. Future studies should include a comprehensive assessment of energy fluxes and potential corrections to avoid spurious energy imbalances. We have included in our conclusion that coupling this model to an ocean state requires that the fluxes are represented well, and that work is underway to address what can be done to tune the model in that way. Additionally, we cite a study which has conducted a supermodel on coupled fluxes and mention the findings in their work (see, Duane and Shen [2023], Shen et al. [2016]).

See Line 341 for the added text:

A key consideration for future work is assessing how the supermodel maintains physical consistency in terms of energy conservation. While our current analysis has focused primarily on wind and temperature fields, a thorough evaluation of radiative fluxes, surface turbulent fluxes, and the overall energy budget will be essential before extending this approach to coupled Earth system models. Ensuring an accurate energy balance will be crucial for improving the fidelity of the supermodel and avoiding unintended biases in simulations. A potential promising avenue of research would be to dynamically connect the fluxes in our supermodeling state as in Shen et al. [2017] which could ensure that each model's energy fluxes are accounted for in the supermodeling framework.

**3. Modes of Variability in the Tropics**

We appreciate the reviewer's suggestion to analyze additional modes of variability beyond PNA and NAO, particularly in the tropics. Given that tropical variability is often strongly influenced by high-frequency convective processes (such as the Madden-Julian Oscillation (MJO) and convectively coupled equatorial waves), a deeper assessment of these features within the supermodel framework would indeed be valuable. We find this analysis potentially beyond the scope of the current manuscript, however, we include a discussion of this point in the manuscript to make the readers aware of the potential issue. We anticipate that due to the large scale nature of the MJO that the synchronization actually might be quite high. Alternatively, it is possible that requiring higher frequency coupling could more directly address this point, this is a point to explore in future work.

Our current results indicate that tropical high-frequency variability is somewhat damped, which may have

implications for the representation of these modes. Investigating how the supermodel handles MJO dynamics, equatorial waves, and other modes of tropical variability is an important area for future research. This could involve spectral analysis of convective variability or direct comparisons with observed MJO characteristics.

See Line 348 for the added text:

> Additionally, our study has primarily examined large-scale variability modes such as the PNA and NAO. However, given the noted reduction in high-frequency variability over the tropics, it will be important to assess the impact on phenomena such as the Madden-Julian Oscillation (MJO) and convectively coupled equatorial waves. These modes play a key role in tropical variability and global teleconnections, and their representation within the supermodel framework remains an important avenue for future research.

**Final Remarks**

While our current study has focused on demonstrating the technical feasibility of the CAM5/CAM6 supermodel and confirming basic synchronization behavior, we agree that a deeper assessment of synchronization patterns, energy budgets, and tropical variability is necessary to fully evaluate the approach. These questions provide a roadmap for future work, particularly as we move toward trained and dynamically weighted supermodels.

We thank the reviewer for these constructive suggestions, which will help guide the next stages of supermodel development.

**4.2. Line Specific Comments**

**RC:** *p2, l28: the NMME and CMIP acronyms need to be defined.*

AR: Fixed, thank you.

**RC:** *p3, l60: typo: one of the two 'to be' needs to be removed.*

AR: Fixed, thank you.

**RC:** *p3, l80: what does 'reference' stand for here? Did you forget to add a reference here?*

AR: Thank you for this comment, yes we had intended to place a reference to our previous manuscript which leveraged the nudging toolbox. It is now in place.

**RC:** *p3, l81: I guess it is 'component models' rather than 'components model'.*

AR: We agree, thank you, it has been fixed.

**RC:** *p4, l123-124: do you mean that sea surface temperatures are constant over each day in CAM (there is thus a small jump at the end of each day)?*

AR: No, we are sorry for the confusion, each time-step has an independent SST field which is smoothly interpolated from time-step to time-step between monthly values.

> which are linearly interpolated to obtain specified independent values at each time-step , as well as the evolution of aerosol emissions and trace gas concentrations (including $CO_2$).

**RC:** *p5, l127: why using a bilinear interpolation and not a conservative one? At least for precipitation, a conservative interpolation sounds more appropriate. Besides, what is the resolution of ERA5 and GPCP datasets?*

**AR:** We appreciate the comment and have switched the precipitation to a conservative regridding and have updated the text. This altered Figure 7 and Figure 9 in the manuscript (slightly) though we note that the results have not changed due to this shift. Additionally, we have added the original resolution of the fields.

> We verify the model against the $\sim0.25°$ ERA5 reanalysis product hersbach2020era5 for all fields except precipitation, which is verified against the $1°$ NOAA GPCP product Adler2003. For verification,  the ERA5 product is bi-linearly interpolated to the native CAM grid prior to any metric calculation. The GPCP product is regridded to the native CAM grid using a conservative mapping method.

**RC:** *p5, l151-154: I feel this technical development requires a bit more explanation to more fully understand how you overcome this challenge of submitting jobs through a single PBS/SLURM scheduler.*

**AR:** We apologize, we have updated how the challenge is handled.

>
>
> To address the challenge of submitting multiple jobs through a single PBS/SLURM scheduler, we implemented a batch submission script that allows two model simulations to be launched concurrently while managing resource allocation efficiently. Our approach ensures that both models utilize the available compute nodes without interfering with each other, thus avoiding scenarios where one model monopolizes the queue while the other remains pending.
>
> Specifically, our script:
>
> 1. **Prepares model runs** by creating initialization files for both simulations.
>
> 2. **Defines model-specific execution settings**, including the number of processing elements required for each job.
>
> 3. **Partitions compute resources dynamically** by selecting appropriate node allocations from `$PBS_NODEFILE`, ensuring that both jobs receive the necessary resources without conflicts.
>
> 4. **Executes model runs in parallel** using background processes (`&`), allowing both jobs to start simultaneously while still being managed within a single job submission.
>
> 5. **Waits for all processes to complete** using `wait`, ensuring that computational resources are fully utilized before job completion.

> This method ensures efficient job scheduling and mitigates the risk of asynchronous queuing delays, ultimately reducing computational time.

**RC:** *Section 2.4: I am a bit confused about how the nudging is performed. Do you average the instantaneous state of the atmosphere over the two model, and then use it for nudging over the 6 following hours (thus the fields toward which the model is nudged are constant over the 6-hour window)? Besides, which nudging timescale to you use?*

**AR:** We have clarified this portion of the text significantly. We encourage the reader to see this section, and include it below to address the comment directly.

> At the beginning of the physics timestep, the first component model outputs the model state variables—Zonal wind ($U$), Meridional wind ($V$), Temperature ($T$), and Specific Humidity ($Q$)—and initiates a model pause by writing a `PAUSE` file. Subsequently, CAM calls a Python script that waits for the second model to reach the beginning of its physics timestep and then combines the outputs from both models at the same timestamp. If the component model grids differ, Python interpolation routines are invoked to ensure consistency. Once the output has been processed, the Python script removes the `PAUSE` file, allowing the model to resume operation without the need for re-initialization or re-entering the queue.
>
> A nudging tendency is then applied to each component model, following Equation (1), which nudges the model state toward the combined model state ($X_{\text{combined}}$) during the first timestep after the models resume running [e.g.,][]Chapman and Berner [2024]. The user can set the nudging timescale ($\tau$); in this experiment, we use a relaxation timescale of 6 hours. Though we emphasize that the tendency is only applied at the first timestep after the combination.
>
> $$\frac{dX}{dt} = F(X) + \frac{X_{\text{combined}} - X}{\tau} \tag{1}$$
>
> where $X$ is the model state, $F(X)$ represents the model's internal tendencies, and $\tau$ is the nudging relaxation timescale.
>
> To address the challenge of submitting multiple jobs through a single PBS/SLURM scheduler, we implemented a batch submission script that allows two model simulations to be launched concurrently while managing resource allocation efficiently. Our approach ensures that both models utilize the available compute nodes without interfering with each other, thus avoiding scenarios where one model monopolizes the queue while the other remains pending.

**RC:** *p6, l167-168: while being an important effort toward open science, this sentence does not seem to be at the right place in these technical description.*

**AR:** We have removed this comment

> The CAM5/CAM6 SuperModel, including the CESM Fortran-Python-bridge, supermodel module toolbox with namelist section, and scheduler scripts, are readily available via the Github Repository.  It is deployed on two HPC platforms 1) the National Center for Atmospheric Research's Derecho Computer and 2) the Norwegian Research Infrastructure Services'

> machine Betzy.

**RC:** *p7, l171-172: this would be interesting to have the elapsed (integrated also) time also for CAM5 and CAM6, to document the overloading of the model synchronisation.*

**AR:** We have added this language to the manuscript.

> ... increase the wallclock time. We acknowledge that this is a significant slowdown from a CAM5/CAM6 simulation which can accomplish a year long simulation in $\sim 2.5$ hours with identical computational resources.

**RC:** *p7, l177: can you elaborate a bit more on this difficulty when adding specific humidity in the nudged state variables?*

**AR:** The main challenges with including specific humidity (Q) in the nudging process stem from the intrinsic properties of moisture in the atmosphere and its coupling with cloud and precipitation processes. In our experience—and as noted in previous studies—several issues arise when directly adjusting Q. Because of these factors, prior work (e.g., Chapman et al. 2024; Raeder et al. 2021) has shown that attempts to nudge Q can introduce more problems than they solve. By allowing Q to evolve freely, we avoid these complications while still effectively constraining the dynamical fields (U, V, and T) that drive the large-scale circulation and, indirectly, the moisture distribution. These considerations motivated our decision to exclude Q from the direct nudging process in the CAM5/CAM6 supermodel implementation. Although this approach may not be entirely satisfying from a theoretical standpoint, it provides a pragmatic solution that leads to a more stable and realistic mean state representation.

We have added a sentence to address this in the manuscript:

> We now demonstrate the synchronization and resulting mean state representation for the CAM5/CAM6 supermodel for the period 1979 through 2005, The supermodel uses an interaction timescale of $\eta = 6$ hours and employs snapshot nudging to the unweighted averaged state. In this implementation, the information in U, V, and T are exchanged and nudged while Q is left to evolve freely. We speculate that the main challenges with including specific humidity (Q) in the nudging process stem from the intrinsic properties of moisture in the atmosphere and its coupling with cloud and precipitation processes.

**RC:** *p8, l195: I guess your refer to Figure 3.*

**AR:** Thank you! This has been adjusted.

**RC:** *p8, l196: 'Fig.' is missing before '3'.a*

**AR:** Thank you! This has been adjusted.

**RC:** *p8, l199-200: without any more detailed analysis, I would argue that the whole atmospheric physics might be at play (most of it is strongly different between CAM5 and CAM6). Besides because the U, V and T forcing in the tropics is in general weaker than in the extratropics, this is rather expected that the model are more sensitive to their own physics, isn't it?*

**AR:** Thank you for your comment, we have chosen to remove this sentence to reflect the uncertainty that you describe.

> Synchronization is strong in the U, V, and T fields poleward of 15, especially in the storm track regions. The Maritime continent region (Lat: [15S,15N], Lon:[60E,200W]) displays the least amount of synchronization .

**RC:** *15. p8, l202: missing ending bracket.*

**AR:** Thank you! This has been corrected.

**RC:** *16. p8, l202-203: The link between the two parts of the sentence remains unclear, and not obviously consistent with what you write l199-200.*

**AR:** We were missing a period, and that has been added, we also clarified our meaning about the pressure levels/

> The supplemental material shows the same analysis for pressure levels of 750 hPa and 900 hPa (Fig. 1S and Fig. 2S). Generally, we see greater synchronization at of U,V, & T at higher pressure levels, while Q has a greater synchronization nearer to the surface, which could be a result of a similar sea surface temperature field.

**RC:** *p8, l207-208: the link is interesting, but probably hard to fully understand for most readers. A bit more explanation would be welcome.*

**AR:** we have opted to remove the sentence to not confuse readers:

>

**RC:** *p8, l218: missing closing bracket.*

**AR:** Thank you! This has been corrected.

**RC:** *p10, l222: do you mean increasing or reducing the relaxation timescale?*

**AR:** We have clarified our meaning

>  In nudging studies, moving to an observations frequency of less than 6 hours seems to alleviate effects of damping davis2022specified, so we hypothesize that increasing the SUMO interaction frequency will be beneficial with regard to minimizing the damping of high frequency variability.

**RC:** *p12, l251: do you mean the correlation between the PNA pattern of SUMO5 and that of SUMO6?*

**AR:** We apologize for the confusion this sentence caused, you are indeed correct. We have amended the language.

>  The principal components corresponding to the SUMO5 and SUMO6 PNA exhibit a Pearson correlation

coefficient of 0.992, whereas the correlation between the CAM6 PNA PC and the SUMO6 PC is only 0.27.

**RC:** *p12, l255: do you have an interpretation or an hypothesis for such a different result between PNA and NAO? Has it been seen in previous studies?*

AR: We do not, likely the NAO represents a weaker signal more greatly affected by internal variability, rather than surface forcing and thus. However, we choose not to speculate in the manuscript.

**RC:** *p13, l258: I guess it should be 'here', not 'her'.*

AR: Thank you! This has been corrected.

**RC:** *p13, l267-268: My understanding is that Figure 4S is showing the differences between the non-interacting ensemble and the supermodel ensemble. It does not seem to correspond to what you are referring to here.*

AR: This figure was incorrectly linked in the Supplemental and has now been corrected and the correct figure is linked.

**RC:** *p16, l289: I would remove the 'in' before CESM.*

AR: Thank you! This has been corrected.

**RC:** *p17, Code and data availability: a recap about the CAM versions and the place where to find the code would be welcome in this section.*

AR: Thank you! The link has been added!

**References**

William E Chapman and Judith Berner. Deterministic and stochastic tendency adjustments derived from data assimilation and nudging. *Quarterly Journal of the Royal Meteorological Society*, 150(760):1420–1446, 2024.

F Counillon, N-S Keenlyside, Shuo Wang, M Devilliers, Alok Gupta, Shunya Koseki, and M-L Shen. Framework for an ocean-connected supermodel of the Earth System. *JAMES*, 15, 2023. .

Gregory S Duane and Mao-Lin Shen. Synchronization of alternative models in a supermodel and the learning of critical behavior. *Journal of the Atmospheric Sciences*, 80(6):1565–1584, 2023.

F. J. Schevenhoven and F. M. Selten. An efficient training scheme for supermodels. *Earth System Dynamics*, 8(2):429–438, 2017. . URL https://www.earth-syst-dynam.net/8/429/2017/.

Francine Schevenhoven. Training of supermodels - in the context of weather and climate forecasting, doctoral thesis. *University of Bergen*, 02 2021. .

Francine Schevenhoven, Frank Selten, Alberto Carrassi, and Noel Keenlyside. Improving weather and climate predictions by training of supermodels. *Earth System Dynamics*, 10(4):789–807, 2019.

C. A. Severijns and W. Hazeleger. The efficient global primitive equation climate model speedo v2.0. *Geoscientific Model Development*, 3(1):105–122, 2010. . URL http://www.geosci-model-dev.net/3/105/2010/.

Mao-Lin Shen, Noel Keenlyside, Frank Selten, Wim Wiegerinck, and Gregory S Duane. Dynamically combining climate models to "supermodel" the tropical pacific. *Geophysical Research Letters*, 43(1): 359–366, 2016.

Mao-Lin Shen, Noel Keenlyside, Bhuwan C Bhatt, and Gregory S Duane. Role of atmosphere-ocean interactions in supermodeling the tropical pacific climate. *Chaos: An Interdisciplinary Journal of Nonlinear Science*, 27(12), 2017.

Oliver Watt-Meyer, Gideon Dresdner, Jeremy McGibbon, Spencer K Clark, Brian Henn, James Duncan, Noah D Brenowitz, Karthik Kashinath, Michael S Pritchard, Boris Bonev, et al. Ace: A fast, skillful learned global atmospheric model for climate prediction. *arXiv preprint arXiv:2310.02074*, 2023.